# Beyond Single-View Indexing: Structure-Aware Multi-View Retrieval for Knowledge-Based VQA

**Hao Wang** [1]  **Xujia Li** [2]  **Lei Chen** [1,2]

## Abstract

Knowledge-Based Visual Question Answering (KB-VQA) relies on retrieval from large-scale knowledge bases, yet this stage is often treated simplistically. Existing methods typically adopt single-view indexing or naive multi-view fusion, leading to systematic coverage gaps. In this work, we demonstrate that different views exhibit strong complementarity in retrieval. Motivated by this observation, we propose SCAR, a Structure-aware Cross-View Retrieval framework that exploits cross-view structural complementarity at inference time without additional training. SCAR enhances retrieval via structure-aware similarity propagation within each view and explicit cross-view redundancy regulation. Experiments on multiple KB-VQA benchmarks demonstrate that SCAR substantially improves retrieval recall, approaches retrieval coverage upper bounds, and consistently boosts end-to-end KB-VQA performance with negligible inference overhead[1].

## 1. Introduction

With the rapid advancement of Multimodal Large Language Models (MLLMs) (Yang et al., 2025a; Wang et al., 2025), recent works have demonstrated strong capabilities in jointly understanding visual and textual information, making them a convincing backbone for Visual Question Answering (VQA) tasks (Caffagni et al., 2024). However, real-world VQA scenarios require access to factual and encyclopedic knowledge that is not explicitly present in the visual input. This is amplified in Knowledge-Based VQA (KB-VQA), where answering questions often depends on external facts

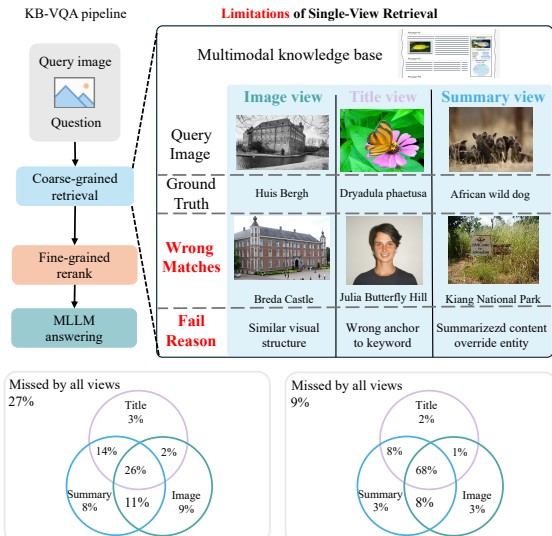

*Figure 1.* View-specific failure examples and Recall@20 overlap of Image, Title, and Summary view retrieval on E-VQA (left) and Infoseek (right). Each circle denotes retrieval results from a single view, with overlaps indicating shared hits across views and non-overlapping regions reflecting view-specific retrieval. **This preliminary evidence illustrates coverage differences across views**.

beyond given visual content. To address this challenge, multi-modal Retrieval-Augmented Generation (mRAG) has emerged as an efficient paradigm for KB-VQA, enabling models to retrieve relevant external knowledge from large-scale knowledge base and ground their answers in explicit evidence (Caffagni et al., 2024).

Advanced mRAG-based KB-VQA systems as shown in Figure 1 follow a general pipeline consisting of coarse-grained retrieval, fine-grained reranking, and MLLM answering (Yang et al., 2025b; Yan & Xie, 2024; Cocchi et al., 2025). While prior work has explored improvements by leveraging stronger rerankers or MLLMs (Yang et al., 2025b; Yuan et al., 2025), the coarse-grained retrieval stage is often treated in a relatively simplistic manner. In particular, existing approaches typically construct the entity index based on a single knowledge view such as using the entity image (Yan & Xie, 2024) or a textual description (Cocchi et al., 2025), or combine multiple such views through heuris-

[1]Data Science and Analytics, Information Hub, HKUST(GZ), Guangzhou, China [2]HKUST, Hong Kong, China. Correspondence to: Xujia Li <leexujia@ust.hk>.

*Proceedings of the $43^{rd}$ International Conference on Machine Learning*, Seoul, South Korea. PMLR 306, 2026. Copyright 2026 by the author(s).

[1]Code implementation link available at: `https://github.com/seraveea/SCAR`

tic score fusion (Zhang et al., 2024). However, we observe that the effectiveness of a given retrieval view varies substantially across datasets (Cocchi et al., 2025). To further investigate this phenomenon, we conduct a preliminary analysis by constructing multiple indexes using different entity elements from a multi-modal knowledge base, following retrieval designs in prior work: a title-based view(Cocchi et al., 2025), an image-based view(Yan & Xie, 2024), and a summary-based view(Yang et al., 2025b). We then evaluate their recall performance on common-used datasets, E-VQA and InfoSeek. We visualize the overlap of retrieved entities using Venn diagrams in Figure 1, which reveal that **each view exhibits complementary strengths**. Beyond statistics, we further show some representative failure cases for each view. As illustrated in Figure 1, these cases highlight that **single-view index has blind spots, leading to coverage gaps at the retrieval stage**. Such limitations have far-reaching implications for KB-VQA, as the effectiveness of downstream re-ranking and answering is fundamentally bounded by coarse-grained retrieval results.

These observations naturally raise a fundamental question: **What influence the effectiveness of retrieval in KB-VQA and how to utilize multi-view from a multi-modal KB?** To explore this, we move beyond empirical observation and provide a theoretical analysis of retrieval from an information-theoretic perspective. Our analysis shows that single-view retrieval is inherently limited by the view-specific mutual information density. More importantly, the potential gain of multi-view retrieval critically depends on the amount of non-redundant information contributed by different views. This analysis motivates us to explore whether multiple retrieval views can improve the recall of coarse-grained retrieval in KB-VQA.

These observations and theoretical insights reveal two key challenges in multi-view retrieval for KB-VQA. Single-view retrieval is limited by view-specific noise and information density, while naive multi-view aggregation often suffers from severe cross-view redundancy, diminishing the benefit of complementary signals. Start from these challenges, we propose **SCAR**, **S**tructure-aware **C**ross-view **A**lignment for multi-modal **R**etrieval. SCAR explicitly models each view-specific index as a semantic structure and performs intra-view manifold propagation to enhance retrieval robustness beyond isolated query–entity similarities. It further coordinates multiple views through structure-aware redundancy regulation, allowing complementary information to be aggregated while suppressing structurally overlapping evidence, thereby strengthening coarse-grained retrieval. SCAR substantially improves retrieval quality, achieving significant gains in recall and consistently approaching the oracle upper bound derived from multi view retrieval. Despite its effectiveness, **SCAR is lightweight and efficient**: SCAR operates entirely at inference time, requires no ad-ditional model training, and introduces only negligible per-query latency overhead.

We summarize contributions as threefold:

- **We identify and empirically demonstrate strong view complementarity at the coarse-grained retrieval stage in KB-VQA**, revealing systematic coverage gaps of single-view retrieval that fundamentally bound downstream performance.

- **We propose SCAR, a training-free and structure-aware multi-view retrieval method**, which improves coarse-grained retrieval by jointly enhancing intra-view semantic structure and regulating cross-view redundancy, without introducing additional models or supervision.

- **Extensive experiments on multiple KB-VQA benchmarks show that SCAR substantially improves retrieval recall**, approaches coverage upper bounds, and integrates seamlessly with existing re-ranking model and MLLMs with negligible inference overhead.

## 2. Related Work

**Knowledge-based visual question answering.** KB-VQA expands the traditional VQA by incorporating external knowledge to answer questions that beyond vision understanding. Datasets like OK-VQA (Marino et al., 2019) and A-OKVQA (Schwenk et al., 2022) contains questions that requiring out-source knowledge to answer. The emergence of E-VQA and InfoSeek (Mensink et al., 2023; Chen et al., 2023; Hung et al., 2025) brings new challenges to KB-VQA task by introducing multi-modal knowledge base. To adopt MLLM in KB-VQA, multi-modal RAG was proposed as a promising direction (Adjali et al., 2024). (Caffagni et al., 2024) was the first work that integrate an external knowledge source in KB-VQA. Many work aimed to fine-tune a MLLM or interact with it (Tian et al., 2025) to enhance its ability of understanding knowledge. By fine-tuning MLLMs, (Qi et al., 2024) was robust to noise introduced by the retrieved multi-modal knowledge. (Cocchi et al., 2025) and (Zhang et al., 2024) finetune the MLLM to let it learn how to make a decision whether the retrieved document is helpful. Another wide-explored direction is to training additional models as re-rankers, to improve knowledge quality and relevance based on the retrieval result (Luo et al., 2025; Deng et al., 2025; Lerner et al., 2024). (Jian et al., 2024) utilize MLLM to extract important entity from query image and perform token-level similarity retrieval. (Yan & Xie, 2024) trained a re-ranker based on retrieved documents according to textual modality. (Yang et al., 2025b) propose a multi-granularity cross-modal retrieval, which boost the retrieval and VQA performance by three-step filtering.

**Retrieval in KB-VQA.** Previous approaches consistently

improve the KB-VQA performance, they lack a systematic discussion of the role of coarse-grained retrieval, which constraint downstream VQA task. (Yan & Xie, 2024) and (Tian et al., 2025) utilize visual signal as anchor to retrieval entities. (Cocchi et al., 2025) found that visual retrieval works well on E-VQA while textual retrieval is better on InfoSeek. (Yang et al., 2025b) employ LLMs to summary entities to obtain macro-level information. (Yuan et al., 2025) run coarse-grained retrieval and then imply proposed graph-based retrieval on the top-10 evidence. (Zhang et al., 2024) use the average of similarity of entity image and text to query image as the retrieval anchor. (Dong et al., 2024) showed that incorporating document structure and textual representations during indexing lead to effective retrieval in textual QA. As shown in Introduction, single views have blind spots, and this can hardly be solved by fine-tuning MLLMs or re-rankers.

## 3. Preliminaries

In this section, we first give definition of the KB-VQA task and formulate the multi-view retrieval setting in this task.

**KB-VQA Task.** Given a query $q$ containing a textual question and an image, an MLLM is expected to generate an answer $y$ by leveraging entity $e$ retrieved from an external knowledge base $\mathcal{E}$ as context. The objective of KB-VQA can therefore be written as

$$y = \arg\max_y \ \mathrm{MLLM}(y \mid q, e). \tag{1}$$

The external knowledge base $\mathcal{E} = \{e_i\}_{i=1}^N$, consists of a collection of knowledge entities. Each entity $e_i$ could be represented by multiple views:

$$e_i = \{x_i^{(j)}\}_{j=1}^M, \tag{2}$$

where view $j$ corresponds to a specific modality or semantic perspective and $x_i^{(j)}$ represents the knowledge of entity $e_i$ in this view. In conventional single-view retrieval, a multimodal encoder $E$ is employed to encode the query $q$ and single view entity representations $x_i$ into a shared embedding space. A coarse-grained retrieval function $F$ returns a set of top-$k$ candidate entities:

$$\mathcal{E}_k = F\big(E(q), \{E(x_i)\}_{i=1}^N, k\big). \tag{3}$$

**Multiview Indexing Retrieval.** As discussed earlier, an entity $e_i$ can be represented by multiple views $\{x_i^{(j)}\}_{j=1}^M$, each leading to a distinct embedding. Instead of a single entity representation $E(e_i)$, we explicitly encode each view and perform retrieval over view-level embeddings. Under this formulation, Eq. 3 can be instantiated as:

$$\mathcal{E}_k = F\big(E(q), \{\{E(x_i^{(j)})\}_{i=1}^N\}_{j=1}^M, k\big). \tag{4}$$

This formulation enables us to exploit multiple knowledge views of each entity at the retrieval stage.

## 4. Theoretical Analysis

Coarse-grained retrieval can be abstracted as a noisy selection over a large knowledge base, where the goal is to include the target entity within a top-$k$ candidate set based on learned representations. In this section, we provide a bound analysis from a theoretical perspective.

**Theorem 4.1** (Single-view Retrieval Upper Bound). *Consider a single view $m$ with ground-truth entity $e$ and view-specific representation $x^{(m)}$. Let $E(\cdot)$ denote the corresponding encoder, and let $k$ be the retrieval cutoff of the retrieval stage. Then, the achievable Recall@K of coarse retrieval under view $m$, denoted by $p_m$, is upper bounded as*

$$p_m \leq \frac{I^{(m)}(x_e^{(m)}; E(x_e^{(m)})) + I(e; E(q)) + \delta_{q,e}^{(m)} + 1}{H_m - \log_2 k}, \tag{5}$$

*where $H_m$ denotes the view-conditioned effective uncertainty from the entity set $\mathcal{E}$ under view $m$, and $I^{(m)}(\cdot)$ denotes the mutual information measured within view $m$. $\delta_{q,e}^{(m)}$ denote the information gain about entity $e$ that introduced by query in view $m$.*

This bound admits a simple interpretation: (i) informative view-specific representations improve the recall by increasing mutual information $I^{(m)}(\cdot)$ and; (ii) the view with higher intrinsic entropy $H_m$ is inherently harder to retrieve from. The full derivation is provided in Appendix A. Based on the analysis in Eq 5, we identify first challenge in coarse-grained retrieval for KB-VQA:

**Challenge 1. Single-view retrieval effectiveness is constrained by how much view-specific mutual information is preserved in the embedding space**. In practice, view-specific noise often distort the similarity (return Mr./Ms. "Butterfly" when retrieving insects using title based index), causing spurious matches that are not supported by true semantic relevance.

**Proposition 4.2** (Multi-view Retrieval Upper Bound). *We consider a multi-view retriever that jointly accesses $M$ view-specific indexes and returns a top-k candidate set, and denote the oracle Recall@k by $p_{ora}$. Following the analysis Theorem 4.1, we adopt an effective-modality approximation to account for cross-view redundancy. The oracle recall admits the following upper bound*

$$p_{ora} \leq \frac{\frac{M_{eff}}{M}\sum_{j=1}^M I^{(j)}(\cdot) + I(e; E(q)) + \sum_{j=1}^M \delta_{q,e}^{(j)} - \kappa\Delta H + 1}{H - \log_2 k}, \tag{6}$$

*where $I^{(j)}(\cdot)$ denotes the mutual information captured under view $j$ as defined in Eq. 5, $H$ denotes the aggregate view-conditioned effective uncertainty across all views, $M_{\mathrm{eff}}$ represents the effective number of non-redundant views, and*

$\kappa\Delta H$ *captures the information overlap induced by structural redundancy among views.* $\sum_{j=1}^{M} \delta_{q,e}^{(j)}$ *denote the maximum query-induced identity information across views.*

Compared to the single-view bound, Eq. 6 introduces two multi-view specific terms. $M_{\text{eff}} = 1 + (M-1)(1-\rho)$ denotes the effective number of independent views, which discounts the nominal view count $M$ by cross-modal redundancy $\rho$. The term $\kappa\Delta H$ captures entropy imbalance across views with different modalities, where $\Delta H$ measures deviation from balanced uncertainty and $\kappa = \frac{1-\rho}{M-1}$ modulates its impact as in (Chen, 2025). The full derivation is provided in Appendix A. Based on Eq.6, we identify the second challenges from multi-view perspective:

**Challenge 2. The benefit of multiple views depends on the amount of non-redundant information they contribute.** Different views may exhibit highly overlapping retrieval results, leading to structural redundancy that reduces the effective number of independent modalities.

## 5. Methodology

Guided by the theoretical insights and empirical observations, we propose a multi-view retrieval method **SCAR: Structure-aware Cross-view Alignment for multi-modal Retrieval**. SCAR improves retrieval by exploiting structural relationships among entities within each view, and coordinating multiple views to avoid redundant evidence. As shown in Figure 2, SCAR consists of two complementary modules, each addressing one of the key challenges identified above. First, **Similarity Propagation via Entity kNN Graphs** tackles the limitation of single-view retrieval by propagating similarity over entity-level neighborhood structures within each view. Second, **Cross-View Redundancy Regulation** addresses the diminishing returns of naive multi-view aggregation by explicitly regulating cross-view contributions, suppressing structurally redundant evidence while preserving complementary information. SCAR operates in a training-free manner, without introducing extra models, and yield efficient performance gains during inference.

### 5.1. Similarity Propagation via Entity kNN Graphs

We first consider reducing noise within each individual view by applying similarity manifold diffusion via entity kNN graphs. For each view $m$, an initial set of coarse retrieval results is obtained as $\mathcal{E}_k^m = F\Big(E(q), \{E(x_i^{(m)})\}, k\Big)$, where $k$ is the number of candidates for each view. A unified entity set is constructed by taking the union of retrieval results across all views, $\mathcal{E}_{\text{union}} = \mathcal{E}_k^1 \cup \mathcal{E}_k^2 \cup \cdots \cup \mathcal{E}_k^M$. For each view $m$, view-specific index vectors of all entities in $\mathcal{E}_{\text{union}}$

are extracted as:

$$\mathbf{x}_i^{(m)} = \begin{cases} E(x_i^{(m)}), & \forall e_i \in \mathcal{E}_k^m, \\ \mathbf{0}, & \text{otherwise.} \end{cases} \quad (7)$$

Based on these representations, an entity kNN graph is constructed for each view $m$. First, Euclidean distances between entities of each view $m$ are computed:

$$d_{ij}^{(m)} = \left\| \frac{\mathbf{x}_i^{(m)}}{\|\mathbf{x}_i^{(m)}\|_2} - \frac{\mathbf{x}_j^{(m)}}{\|\mathbf{x}_j^{(m)}\|_2} \right\|_2. \quad (8)$$

We then construct the graph affinity matrix by applying a Gaussian kernel to $d_{ij}^{(m)}$:

$$W_{ij}^{(m)} = \exp(-\frac{d_{ij}^{(m)} \cdot d_{ij}^{(m)}}{2\sigma^2}), \ \sigma_m = \text{mean}(\{d_{ij}^{(m)}\}), \quad (9)$$

where $\sigma_m$ is a scale parameter that controls the sensitivity of the affinity function under view $m$. Entities absent from view $m$ are excluded from graph propagation by removing all incident edges:

$$W_{i,:}^{(m)} = W_{:,i}^{(m)} = 0, \text{ if } \mathbf{x}_i^{(m)} = 0. \quad (10)$$

This design prevents entities that are not observed in a given view from being artificially influenced by unrelated neighbors. The affinity matrix is sparsified by retaining only the top-$n$ nearest neighbors according to $W^{(m)}$.

Given the entity kNN graph, manifold diffusion is performed to propagate relevance scores along the graph (Premachandran & Kakarala, 2013; Lin et al., 2023). The symmetrically normalized affinity matrix is defined as

$$\mathbf{S}^{(m)} = \mathbf{D}^{(m)-1/2}\mathbf{W}^{(m)}\mathbf{D}^{(m)-1/2}, \ D_{jj}^{(m)} = \sum_k W_{jk}^{(m)}. \quad (11)$$

Manifold propagation is applied to refine relevance scores, admitting a closed-form solution (Iscen et al., 2017; Loy et al., 2013). Given the normalized affinity matrix and the initial relevance vector $\mathbf{y}$, the refined scores for view $m$ are obtained as

$$\mathbf{z}^{(m)} = (\mathbf{I} - \alpha\mathbf{S}^{(m)})^{-1}\mathbf{y}, \quad (12)$$

where $\alpha \in (0,1)$ controls the strength of propagation and $\mathbf{z}$ denotes the smoothed relevance scores. Here, the initial score $\mathbf{y}$ is defined by summing the initial retrieval similarities across all views. This design ensures that differences among views arise solely from the structural properties, rather than from view-specific retrieval scores. The propagation is performed independently for each view. Through this module, entities in a strong neighborhood are jointly reinforced, whereas isolated hallucinated entities receive reduced scores.

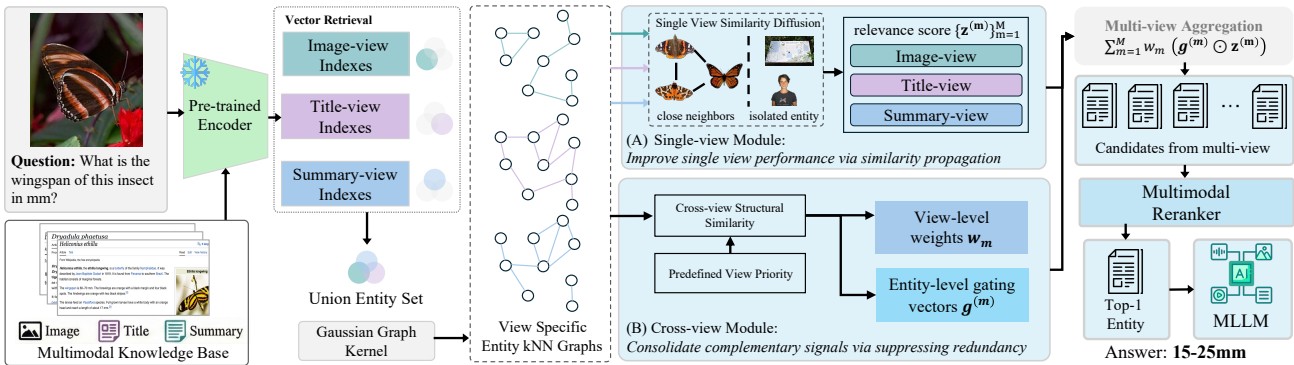

*Figure 2.* SCAR enhances coarse-grained retrieval by propagating similarity over entity-level kNN graphs within each view and regulating cross-view redundancy based on structural overlap, enabling effective aggregation of complementary knowledge views.

## 5.2. Cross-View Redundancy Regulation

While similarity propagation reduces noise within each view, it does not explicitly address redundancy across different views. Multiple views may provide highly correlated or overlapping evidence for the same entities, leading to over-amplification of redundant signals and diminished contributions from other views. To mitigate this issue, we introduce Cross-View Redundancy Regulation, which explicitly detects structural redundancy among similarity propagation results from multiple views, and applies dual-granularity suppression to lower-priority views.

For each view $m$, a view-specific affinity matrix $\mathbf{S}^{(m)}$ has been constructed in Eq. 11. To assess whether two views introduce redundant structural information, we measure the similarity between their structures. Given two views $j_1$ and $j_2$, we consider only the entities that appear in both views and extract the corresponding induced subgraphs:

$$\mathcal{E}_{j_1 j_2} = \{e \mid \mathbf{x}_i^{(j_1)} \neq \mathbf{0} \land \mathbf{x}_i^{(j_2)} \neq \mathbf{0}\}. \quad (13)$$

On the shared entity set $\mathcal{E}_{j_1 j_2}$, we restrict the affinity matrices to obtain $\mathbf{S}_{\mathcal{E}_{j_1 j_2}}^{(j_1)}$ and $\mathbf{S}_{\mathcal{E}_{j_1 j_2}}^{(j_2)}$, where $\mathbf{S}_{\mathcal{E}_{j_1 j_2}}^{(\cdot)}$ denotes the submatrix obtained by restricting rows and columns to indices in $\mathcal{E}_{j_1 j_2}$. The structural similarity between the two subgraph is then computed:

$$\text{sim}(j_1, j_2) = \frac{\langle \mathbf{S}_{\mathcal{E}_{j_1 j_2}}^{(j_1)}, \mathbf{S}_{\mathcal{E}_{j_1 j_2}}^{(j_2)} \rangle_F}{\|\mathbf{S}_{\mathcal{E}_{j_1 j_2}}^{(j_1)}\|_F \cdot \|\mathbf{S}_{\mathcal{E}_{j_1 j_2}}^{(j_2)}\|_F}, \quad (14)$$

where $\langle \cdot, \cdot \rangle_F$ denotes the Frobenius inner product. Large $\text{sim}(j_1, j_2)$ indicates that views $j_1$ and $j_2$ are considered to exhibit significant structural redundancy. When there are more than two views, the above equation is applied pairwise across all view pairs. Directly aggregating their diffusion scores would therefore repeatedly amplify the same entities. To avoid such symmetric reinforcement, we conduct a dual-granularity regulation. First, a priority ordering over views is predefined, $\pi : \{1, \dots, M\} \to \mathbb{R}$, where a lower value of

$\pi(m)$ indicates a higher-priority view, reflecting its expected contribution to multi-view retrieval. When $\text{sim}(j_1, j_2)$ is larger than a threshold $\tau$, the lower-priority view is first subjected to a view-level global scaling to reduce its overall influence in the final retrieval score:

$$\omega_{j_{\text{low}}} \leftarrow \gamma \omega_u, \qquad \gamma \in (0, 1), \quad (15)$$

where $j_{low}$ denote the view that have lower-priority and the view-level weight $\omega_u$ is uniform initialized.

Subsequently, a stronger entity-level suppression is applied only to the overlapping entities shared by the two views. Specifically, for each view $m$, an entity-level gating vector $\boldsymbol{g}^{(m)} \in \mathbb{R}^{|\mathcal{E}_{\text{union}}|}$ is defined, where each element $g_i^{(m)}$ modulates the contribution of entity $e_i$ under view $m$. The gating vector is initialized as $g_i^{(m)} = 1$ for all entities observed in view $m$ and $g_i^{(m)} = 0$ otherwise. When structural redundancy is detected between two views, $g_i^{(m)}$ in low-priority view is further down-scaled by a factor $\beta \in (0, 1)$ for entities belonging to the overlapping entity set. The final cross-view relevance score is computed as

$$\mathbf{z}_{\text{final}} = \sum_{m=1}^{M} w_m (\boldsymbol{g}^{(m)} \odot \mathbf{z}^{(m)}), \quad (16)$$

where $w_m$ denotes the view-level weight and $\mathbf{z}^{(m)}$ denotes the normalized diffusion scores of view $m$. The top-$k$ entities ranked by $\mathbf{z}_{\text{final}}$ are selected as the retrieval candidates.

## 6. Experiments

**Datasets.** We evaluate SCAR on KB-VQA benchmarks, E-VQA (Mensink et al., 2023) and InfoSeek (Chen et al., 2023). For E-VQA, we use the knowledge base provided by the dataset. For InfoSeek, to be consistent with existing methods, we follow the settings of (Yan & Xie, 2024), using the 100k knowledge base. For each dataset, we employ three view-specific indexes: an image-based index (Yan & Xie, 2024), a summary-based index (Yang et al., 2025b),

*Table 1.* Retrieval performance on KB-VQA benchmarks. $\diamond$ represents our reproductions. Best results are highlighted in **bold**. SCAR consistently outperforms prior methods and approaches the multi-view coverage upper bound on both E-VQA and InfoSeek.

| Method | Encoder | E-VQA | | | | InfoSeek | | | |
|---|---|---|---|---|---|---|---|---|---|
| | | R@1 | R@5 | R@10 | R@20 | R@1 | R@5 | R@10 | R@20 |
| *Multi-view coverage upper bound* | | *33.9* | *56.9* | *65.8* | *73.0* | *68.9* | *84.2* | *87.9* | *90.7* |
| Wiki-LLaVA (Caffagni et al., 2024) | CLIP ViT-L/14+Contriever | 3.3 | 9.9 | 13.2 | 17.5 | 36.9 | 66.1 | 71.9 | 78.4 |
| mR$^2$AG (Zhang et al., 2024) | CLIP-ViT-L/14@336px | - | - | - | - | 38.0 | 53.0 | 65.0 | - |
| EchoSight (Yan & Xie, 2024) | EVA-CLIP-8B | 13.3 | 31.3 | 41.0 | 48.8 | 45.6 | 67.1 | 73.0 | 77.9 |
| ReflectiVA (Cocchi et al., 2025) | EVA-CLIP-8B | 15.6 | 36.1 | - | 49.8 | 56.1 | 77.6 | - | 86.4 |
| OMGM$^\diamond$ (Yang et al., 2025b) | EVA-CLIP-8B | 18.7 | 41.2 | 49.7 | 58.7 | 52.2 | 73.7 | 79.8 | 84.7 |
| mKG-RAG (Yuan et al., 2025) | QM-Retriever | 18.9 | 36.8 | 46.2 | 55.6 | 49.7 | 71.6 | 78.0 | 82.5 |
| **SCAR**(ours) | EVA-CLIP-8B | **28.8** | **49.9** | **57.0** | **65.1** | **60.8** | **80.0** | **84.6** | **87.7** |
| *SCAR / coverage upper bound(%)* | | 85.96 | 87.70 | 86.62 | 89.18 | 88.24 | 95.01 | 96.25 | 96.69 |

and a title-based index (Cocchi et al., 2025). These indexes correspond to three views considered in multi-view retrieval. We evaluate retrieval quality using standard metrics Recall@K. For VQA performance, we report results using the official evaluation metrics provided by each dataset (BEM score (Zhang et al., 2019) for E-VQA, VQA accuracy (Antol et al., 2015) and relaxed accuracy (Methani et al., 2020) for InfoSeek). More details on the datasets and evaluation methods can be found in the Appendix B.1.

**Implementation Details** To maintain consistency with prior work (Yan & Xie, 2024; Cocchi et al., 2025) and ensure that performance gains stem from the retrieval mechanism rather than a different multi-modal encoder, we use EVA-CLIP-8B in SCAR. Consistent with previous studies (Yan & Xie, 2024; Cocchi et al., 2025), we retrieve the top-20 most relevant entities. In single-view diffusion stage, we set the strength of propagation $\alpha = 0.9$, and choose 10 neighbors to build entity kNN graphs. In the cross view redundancy regulation stage, we set the threshold of regulation as $\tau = 0.7$. The view-level scaling weight $\gamma = 0.8$ and the entity-level down scale factor is set to 0.2. The priority is set to $summary \rightarrow image \rightarrow title$. All hyperparameters are fixed for simplicity and reused across datasets without tuning; we report detailed sensitivity analyses in Section 6.4 and Appendix C.2 for transparency. During the question answering stage, we adopt pre-trained MLLMs as answer generators. All experiments are conducted in a zero-shot manner, and more details as well as the prompts used for answer generation are provided in the Appendix B.2 and B.1.

### 6.1. Experimental Results

The main experimental results of SCAR, together with comparisons to SOTA baselines are presented in Table 1. SCAR consistently outperforms baselines across Recall@K. These results demonstrate the effectiveness of SCAR and highlight the necessity of multi-view indexing for KB-VQA over multi-modal knowledge bases. Single-view index is fundamentally constrained by the information density and

*Table 2.* Ablation study of different view combinations on E-VQA. We report Recall@K for single-view, pairwise combinations, and three-view retrieval settings. ✓indicate that view is included.

| Image | Title | Summary | R@1 | R@5 | R@10 | R@20 |
|---|---|---|---|---|---|---|
| ✓ | | | 13.4 | 31.8 | 41.8 | 48.8 |
| | ✓ | | 17.6 | 31.9 | 38.7 | 44.7 |
| | | ✓ | 18.7 | 41.2 | 49.7 | 58.7 |
| ✓ | ✓ | | 25.4 | 44.3 | 52.0 | 59.4 |
| ✓ | | ✓ | **28.8** | 46.5 | 54.4 | 62.9 |
| | ✓ | ✓ | 22.3 | 38.0 | 46.3 | 54.9 |
| ✓ | ✓ | ✓ | **28.8** | **49.9** | **57.0** | **65.1** |

view-specific noise. In contrast, multi-view retrieval enables mutual verification from different views, increasing the density of query-relevant information during retrieval.

**Approaching the retrieval coverage bound.** We compute the multi-view coverage upper bound of retrieval, which is located in the top of Table 1. Specifically, we define a multi-view retrieval upper bound as follows: if the ground-truth entity appears in the union set $\mathcal{E}_{union}$ of all views where $|\mathcal{E}_{union}| \leq MK$, the recall@K is counted as 1. This upper bound represents the best achievable performance of multi-view indexing under the given encoders. In Table 1 we observe that on E-VQA, SCAR reaches 89.2% of the upper bound at Recall@20, while on InfoSeek it achieves 96.7%. These results indicate that SCAR is able to approximate the multi-view retrieval upper bound, providing strong empirical evidence of its effectiveness. Notably, the gap between our method and the upper bound consistently narrows as $K$ increases from Recall@1 to Recall@20. When expanding the cutoff k, SCAR increasingly approaches the theoretical upper bound, which aligns with the analysis in Eq.6.

**Impact of different views in retrieval.** We conduct studies to analyze the impact of different view combinations in SCAR. Specifically, we evaluate all single views as well as all pairwise combinations of the image, title, and summary views. The retrieval results on E-VQA are reported in Table 2, while the results on Infoseek are provided in the Appendix C due to space limitations. From the results in Table 2, using all three views yields the best retrieval per-

*Table 3.* Retrieval performance on E-VQA with a re-ranking stage. All methods are evaluated using a re-ranking module from OMGM. **SCAR is effective and outperforms baselines with re-ranking**.

| Method | R@1 | R@5 | R@10 | R@20 |
|---|---|---|---|---|
| Title View | 29.6 | 38.9 | 40.0 | 44.7 |
| Image View | 32.6 | 45.5 | 46.8 | 48.9 |
| EchoSight | 36.5 | 47.9 | 48.8 | 48.8 |
| OMGM$^\diamond$ | 39.0 | 51.5 | 53.7 | 58.7 |
| SCAR | **43.1** | **58.1** | **59.7** | **65.1** |

formance. This confirms that incorporating multiple views provides complementary information from different modalities and perspectives, which is beneficial for improving retrieval effectiveness. Among the pairwise combinations, the "image + summary" setting achieves the second-best performance. This can be explained by our analysis in Eq 6. Both image and summary views exhibit strong single-view retrieval performance, and more importantly, they capture complementary information, resulting in a higher $M_{eff}$. In contrast, the "summary + title" combination performs worse than other combinations. Although the summary view alone achieves better retrieval performance, summary and title are both text-based views, leading to a smaller $M_{eff}$ due to higher view redundancy.

**Integrating a Re-ranking stage.** Since re-ranking have become common components in advanced KB-VQA pipelines (Yuan et al., 2025; Hong et al., 2025), we further evaluate SCAR in conjunction with the re-ranking module. We employ a frozen OMGM reranker (Yang et al., 2025b), a Q-Former based model (Li et al., 2023), using the default weights and settings. The results are reported in Table 3, where we compare SCAR against single-view retrieval baselines on the E-VQA dataset. The result of InfoSeek is presented in Appendix C. As shown in Table 3, SCAR consistently achieves significant improvements after re-ranking. These results validate our motivation for optimizing retrieval stage: retrieval quality serves as the foundation of the KB-VQA, and failures at the retrieval stage, where relevant entities are missed, cannot be effectively recovered by re-ranking modules.

**KB-VQA performance.** We further evaluate SCAR on the KB-VQA task using three pre-trained VLLMs in Table 4. These results demonstrate that the improvements brought by SCAR at the retrieval stage translate effectively to downstream QA task. We also include a detailed comparison of SOTA baselines in question-answer in Appendix C.

### 6.2. Efficiency and Computational Cost

We further measure the runtime on both E-VQA and Infoseek under identical experimental settings. All experiments are conducted on a NVIDIA H100 GPU with 80GB memory. We report the average per-query runtime for retrieval,

*Table 4.* KB-VQA performance under different retrieval views. **SCAR consistently outperforms single-view retrieval**. Best results are highlighted in **bold**.

| MLLM | Retrieval | E-VQA | Infoseek | | |
|---|---|---|---|---|---|
| | | All | Unseen-Q | Unseen-E | All |
| LLaVA-1.5-7B | Image | 26.5 | 27.1 | 24.7 | 25.9 |
| | Title | 27.7 | 28.6 | 27.2 | 27.9 |
| | Summary | 28.8 | 29.3 | 28.9 | 29.1 |
| | SCAR | **34.4** | **31.5** | **30.5** | **31.0** |
| Qwen3-VL-8B | Image | 31.0 | 29.9 | 28.4 | 29.1 |
| | Title | 32.1 | 30.5 | 31.0 | 30.7 |
| | Summary | 33.4 | 31.8 | 32.4 | 32.1 |
| | SCAR | **39.8** | **34.1** | **34.9** | **34.5** |
| InternVL3.5-8B | Image | 28.3 | 28.3 | 26.5 | 27.4 |
| | Title | 31.2 | 28.8 | 29.1 | 28.9 |
| | Summary | 32.4 | 30.0 | 30.6 | 30.3 |
| | SCAR | **39.0** | **31.9** | **32.6** | **32.3** |

*Table 5.* Average per-query runtime (ms) on E-VQA and InfoSeek. SCAR yields **substantial recall improvements** by introducing a **marginal overhead**.

| | Retrieval | Rerank | Generation | Total | Recall |
|---|---|---|---|---|---|
| **E-VQA** | | | | | |
| Title | 44.3±30.8 | 7689.4±4173.5 | 713.7±352.6 | 8447.4 | 44.7 |
| Image | 59.0±24.6 | 7750.5±4616.0 | 725.7±464.3 | 8535.2 | 48.8 |
| Summary | 46.4±28.9 | 7780.8±4829.7 | 717.6±261.7 | 8544.8 | 58.7 |
| SCAR | 85.4±40.7 | 7768.5±4154.3 | 714.7±313.9 | 8568.6 | **65.1** |
| **InfoSeek** | | | | | |
| Title | 23.4±13.7 | 5671.7±2430.7 | 694.7±244.3 | 6389.8 | 78.9 |
| Image | 35.5±17.0 | 5584.3±3118.7 | 682.4±256.9 | 6302.2 | 76.0 |
| Summary | 43.7±28.5 | 5693.3±2150.9 | 694.6±244.4 | 6431.6 | 84.7 |
| SCAR | 46.8±16.4 | 5705.8±3472.9 | 685.3±257.3 | 6437.9 | **87.7** |

re-ranking, and generation, as well as the total latency, in Table 5. Due to the structure of the image index, where multiple images may correspond to a single entity, image-based retrieval requires an additional mapping step from images to entities, resulting in longer retrieval time. **Notably, our method achieves substantial improvements in retrieval at the cost of only a marginal increase in latency.** On E-VQA, SCAR introduces approximately 40 ms additional overhead in the retrieval, while delivering significant recall gains. **This overhead accounts for less than 0.5% of the average latency (around 8000 ms per query).** From a space perspective, multi-view retrieval requires multiple indexes into GPU memory. In practice, a FAISS index over 2 million entities occupies about 20GB memory, which fits within current GPU capacities. Overall, SCAR offers a highly favorable efficiency–performance trade-off.

### 6.3. Ablation Study

**Effect of Modules 1&2.** We conduct experiments to analyze the contribution of each component. Table 6 shows that all components contribute positively to the performance. Notably, averaging multi-view initial retrieval similarities leads to inferior performance, even defeated by the best single-view in Table 2. This result suggests that naive multi-

*Table 6.* Ablation study of SCAR modules. ✓indicate the enabled modules. Both modules yield consistent improvements.

| Module I | Module II | R@1 | R@5 | R@10 | R@20 |
|:---:|:---:|:---:|:---:|:---:|:---:|
| | | 24.6 | 42.3 | 46.4 | 56.4 |
| ✓ | | 28.4 | 48.3 | 56.3 | 63.8 |
| ✓ | ✓ | **28.8** | **49.9** | **57.0** | **65.1** |

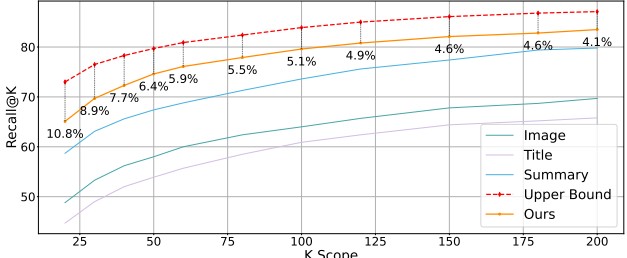

*Figure 3.* Effect of retrieval scope. Increasing from 20 to 200, SCAR consistently improves retrieval performance. The percentages denote the relative gap between SCAR and the upper bound, computed as (UpperBound − SCAR)/UpperBound.

view aggregation is insufficient. We analyze the activation frequency of redundancy regulation and find that it is triggered in approximately 80% of queries, indicating prevalent structural overlap across views in KB-VQA retrieval. For more ablation study and detailed number of regulation triggered in each pair of views, please refer to Appendix C.

**Impact of retrieval scope.** We varying the candidate size from K=20 to K=200 in retrieval. As shown in Figure 3, **SCAR consistently improves across the entire range and remains effective under all retrieval scopes**. The relative gap between SCAR and the upper bound decreases from 10.8% to 4.1%, indicating that SCAR is increasingly approaching the upper bound while K increasing. This behavior is well aligned with our multi-view upper bound analysis: increasing K enlarges the $\log_2 K$ term in the denominator of Eq. 6. The discussion of single-view baseline trend and the average running time is included in Appendix C.1.

### 6.4. Hyperparameter Sensitivity Analysis

**Impact of diffusion strength $\alpha$.** We study the impact of the diffusion strength $\alpha$ in Figure 4. Retrieval improves as $\alpha$ increases, indicating that stronger propagation effectively reinforces semantically coherence. We study the high-propagation regime by fine-grained experiments while $\alpha \in [0.8, 1.0]$. Figure 4 shows that performance remains stable within this interval, with a slight saturation trend as $\alpha$ approaches 1.

**View priority ordering.** We analyze the sensitivity of SCAR to view priority ordering by evaluating all permutations of the three views. As shown in Figure 5, different priority orderings lead to only minor and bounded performance variations, while all permutations consistently outperform

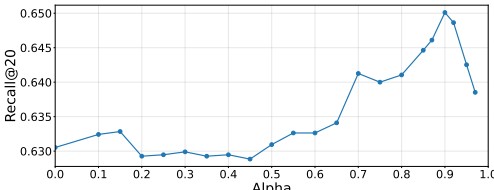

*Figure 4.* Sensitivity analysis of diffusion strength $\alpha$ on E-VQA.

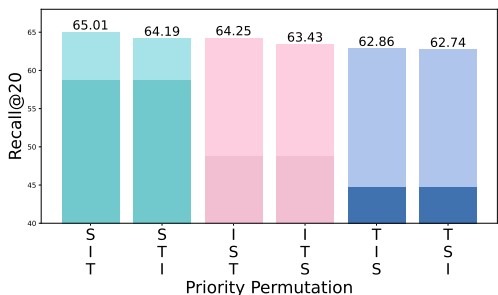

*Figure 5.* Recall@20 under different view priority permutations on E-VQA. I, T, and S denote Image, Title, and Summary. The order indicates decreasing priority. The darker parts correspond to the Recall@20 achieved by the highest-priority single view alone.

their corresponding single-view baselines. This indicates that SCAR is robust to the choice of view priority, and can reliably leverage complementary information even when the highest-priority single view is relatively weak. While assigning higher priority to more informative views yields slightly better results, the overall gains are not dominated by the ordering. A closer inspection suggests that separating structurally similar text-based views (Summary and Title) in the priority list can be beneficial, implying that under high structural redundancy, relative priority distance plays a secondary role compared to multi-view complementarity.

Complete sensitivity analyses of kNN graph size in propagation, regulation-stage hyperparameters, similarity threshold $\tau$, the view-level reduction factor $\gamma$, and the entity-level suppression factor $\beta$, are deferred to Appendix C.2. Across wide ranges of these hyperparameters, SCAR exhibits stable retrieval performance with only marginal variance, indicating that its gains stem from structural modeling rather than parameter tuning. We provide qualitative case studies in Appendix C.3 to illustrate how SCAR integrates complementary signals from different view-specific indexes.

## 7. Conclusions

We revisit coarse-grained retrieval as a key bottleneck in KB-VQA and show that different views provide complementary information. Based on this insight, we propose SCAR, a training-free multi-view retrieval framework that enhances intra-view structure while regulating cross-view redundancy. Empirical results show that SCAR consistently improves recall with negligible overhead and approaches the

multi-view coverage upper bound. Future work may explore more memory-efficient designs for multi-view indexing, including cross-view representation sharing or compression techniques that preserve complementary information. In addition, leveraging lightweight internal signals from MLLMs during retrieval to quickly estimate candidate usefulness could enable more adaptive and efficient retrieval pipelines.

## Impact Statement

This work studies retrieval mechanisms for multi-modal question answering systems. The techniques introduced here may contribute to improving the robustness of retrieval-augmented generation, and their broader societal implications are consistent with those commonly associated with information retrieval and question answering research. As our contributions focus on retrieval-stage coordination without introducing new data sources or deployment-specific assumptions, we do not identify additional ethical concerns that require separate discussion.

## Acknowledgements

Lei Chen's work is partially supported by National Key Research and Development Program of China Grant No. 2023YFF0725100, National Science Foundation of China (NSFC) under Grant No. U22B2060, Guangdong-Hong Kong Technology Innovation Joint Funding Scheme Project No. 2024A0505040012, the Hong Kong RGC GRF Project 16213620, RIF Project R6020-19, AOE Project AoE/E-603/18, Theme-based project TRS T41-603/20R, CRF Project C2004-21G, Key Areas Special Project of Guangdong Provincial Universities 2024ZDZX1006, Guangdong Province Science and Technology Plan Project 2023A0505030011, Guangzhou municipality big data intelligence key lab, 2023A03J0012, Hong Kong ITC ITF grants MHX/078/21 and PRP/004/22FX, Hong Kong ITC TC-SKLCRCC26EG01, Zhujiang scholar program 2021JC02X170, Microsoft Research Asia Collaborative Research Grant, HKUST-Webank joint research lab, 2025 HKUST Shenzhen-Hong Kong Collaborative Innovation Institute Green Sustainability Special Fund from Shui On Xintiandi and the InnoSpace GBA, and HKUST(GZ) - CMCC(Guangzhou Branch) Metaverse Joint Innovation Lab under Grant No. P00659.

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

# Appendix

## A. An information-theoretic interpretation of retrieval limits.

Given a knowledge base $\mathcal{E}$, which we have multiple index system $\{\{E(x_i^{(j)})\}_{i=1}^N\}_{j=1}^M$ that built by different views.

### A.1. Single-view upper bound

First we induce the upper bound of single index Recall@k $p_m$. We note the ground truth as $e$, the knowledge in view $m$ as $x_e^{(m)}$ and the embedding as $E(x_e^{(m)})$. According to information theory and Markov Chain, we have:

$$e \to x_e^{(m)} \to E(x_e^{(m)}). \tag{17}$$

According to the chain rule of mutual information,

$$I^{(m)}\big(x_e^{(m)}, e; E(x_e^{(m)})\big) = I^{(m)}\big(x_e^{(m)}; E(x_e^{(m)})\big) + I^{(m)}(e; E(x_e^{(m)})|x_e^{(m)}). \tag{18}$$

Since the encoder $E(\cdot)$ only takes $x_e^{(m)}$ as input, $E(x_e^{(m)})$ is conditionally independent of $e$ given $x_e^{(m)}$. So the mutual information $I^{(m)}\big(x_e^{(m)}, e; E(x_e^{(m)})\big) = I^{(m)}\big(x_e^{(m)}; E(x_e^{(m)})\big)$. Then we use the chain rule to present $I^{(m)}(x_e^{(m)}, e; E(x_e^{(m)}))$ in another form:

$$I^{(m)}\big(x_e^{(m)}, e; E(x_e^{(m)})\big) = I^{(m)}\big(e; E(x_e^{(m)})\big) + I^{(m)}(x_e^{(m)}; E(x_e^{(m)})|e), \tag{19}$$

combining with $I^{(m)}\big(x_e^{(m)}, e; E(x_e^{(m)})\big) = I^{(m)}\big(x_e^{(m)}; E(x_e^{(m)})\big)$, we have:

$$I^{(m)}\big(x_e^{(m)}; E(x_e^{(m)})\big) = I^{(m)}\big(e; E(x_e^{(m)})\big) + I^{(m)}(x_e^{(m)}; E(x_e^{(m)})|e). \tag{20}$$

Since mutual information is non-negative, we get:

$$I^{(m)}\big(x_e^{(m)}; E(x_e^{(m)})\big) \geq I^{(m)}\big(e; E(x_e^{(m)})\big), \tag{21}$$

$I^{(m)}\big(x_e^{(m)}; E(x_e^{(m)})\big)$ represent the information that we can extract by encoding $x_e^{(m)}$. This inequality indicates that we can only obtain information about the ground truth $e$ by encoding knowledge $x_e^{(m)}$ in view $m$. We adopt a relaxed version of Fano's inequality (Scarlett & Cevher, 2019) by treating top-$k$ retrieval as a set decoding problem:

$$H_m\big(e|F(E(q), E(\{x_i^{(m)}\}), k)\big) \leq h_b(p_m) + p_m \log_2 k + (1 - p_m)H_{rem} \tag{22}$$

where $F(\cdot)$ denotes retrieving the top-$k$ candidates, and $H_{rem}$ is the information entropy that $e$ is not in the retrieved top-$k$ list. For notational simplicity, we write $F(\cdot)$ as a shorthand for $F(E(q), E(\{x_i^{(m)}\}), k)$. According to the definition of mutual information, we have

$$I^{(m)}\big(e; F(\cdot)\big) = H_m(e) - H_m\big(e|F(\cdot)\big), \tag{23}$$

We emphasize that $H_m$ does not denote the prior entropy of the entity, which is view-independent. Instead, $H_m$ captures the view-conditioned effective uncertainty of identifying the target entity under view $m$, defined as $H_m := H(e \mid E(x^{(m)}))$. This quantity reflects how informative view m is about the entity identity after encoding, and may vary across views depending on representation quality and semantic coverage. We use $H_m$ to denote $H_m(e)$. Different views may expose different levels of semantic ambiguity, leading to different entropies $H_m$, even though the underlying entity set remains the same. Since $F(\cdot)$ is a deterministic function of $(E(q), E(\{x_i^{(m)}\}))$, by DPI we have

$$I^{(m)}\big(e; F(\cdot)\big) \leq I^{(m)}\big(e; E(q), E(\{x_i^{(m)}\})\big). \tag{24}$$

For all embedding in the view m, we could split them into two groups: 1. the embedding of the target entity $E(x_e^{(m)})$ and the embeddings of other entities $E(\{x_i^{(m)}\}_{i \neq e})$ Apply the chain rule:

$$I^{(m)}(e; E(x_e^{(m)}), E(\{x_i^{(m)}\}_{i \neq e}) \mid E(q)) = I^{(m)}(e; E(x_e^{(m)}) \mid E(q)) + I^{(m)}(e; E(\{x_i^{(m)}\}_{i \neq e}) \mid E(q), E(x_e^{(m)})) \tag{25}$$

Here we introduce an assumption in KB-VQA retrieval:

**Assumption 1 (Conditional Independence of Distractor Representations).** In the knowledge base $\mathcal{E}$, each entity $e_i$ is associated with a view-specific knowledge representation $x_i^{(m)}$. We assume the following conditional independence property for KB-VQA retrieval: conditioned on the target entity representation $x_e^{(m)}$ and the query embedding $E(q)$, the representations of all non-target entities $\{x_i^{(m)}\}_{i \neq e}$ are statistically independent of the target entity identity $e$:

$$I^{(m)}\Big(e;\ E(\{x_i^{(m)}\}_{i \neq e}) \ \Big| \ E(q),\ E(x_e^{(m)})\Big) = 0. \tag{26}$$

This assumption reflects the retrieval setting where only the target entity's own knowledge representation carries discriminative information about its identity, while the remaining entities act as distractors whose embeddings do not provide additional evidence for identifying e once the target representation is observed.

Assumption 1 is a technical simplification and is not meant to faithfully model modern embedding spaces, where semantically similar entities form clusters. Importantly, KB-VQA supervision typically assigns a unique target entity to each query; thus, such clustering mainly creates hard negatives and increases ambiguity rather than providing helpful evidence. In this sense, dependence among distractors should be viewed as structured noise. Therefore, we interpret the bounds as qualitative, highlighting limiting factors and scaling behavior rather than claiming quantitative tightness.

Then we have:

$$\begin{aligned}
I^{(m)}\big(e; E(q), E(\{x_i^{(m)}\})\big) &= I^{(m)}\big(e; E(q)\big) + I^{(m)}(e; E(\{x_i^{(m)}\}) \mid E(q)) \\
&= I^{(m)}\big(e; E(q)\big) + I^{(m)}(e; E(x_e^{(m)}), E(\{x_i^{(m)}\}_{i \neq e}) \mid E(q)) \\
&= I^{(m)}\big(e; E(q)\big) + I(e; E(x_e^{(m)}) \mid E(q)) \ + \ I(e; E(\{x_i^{(m)}\}_{i \neq e}) \mid E(q), E(x_e^{(m)})) \\
&= I^{(m)}\big(e; E(q)\big) + I^{(m)}(e; E(x_e^{(m)}) \mid E(q))
\end{aligned} \tag{27}$$

While the query is essential for selecting the relevant entity in KB-VQA, its role in conveying entity-identity information is fundamentally different from that of the knowledge base. The passage $x_e^{(m)}$ is designed to encode entity-specific knowledge under view $m$, whereas the query primarily specifies which aspects of the entity are queried, rather than introducing new identity information. Therefore we introduce an assumption to describe the information gain of query:

**Assumption 2 (Bounded query-side identity gain).** In view $m$, the query embedding $E(q)$ may provide additional information about the ground-truth entity $e$ beyond what is already contained in the target knowledge representation $E(x_e^{(m)})$. We assume such additional identity information is bounded:

$$I^{(m)}\big(e;\ E(q) \mid E(x_e^{(m)})\big) \leq \delta_{q,e}^{(m)}, \tag{28}$$

where $\delta_m \geq 0$ is a view-dependent slack term capturing the maximum extra disambiguation signal that the query can contribute after observing $E(x_e^{(m)})$.

Applying the chain rule:

$$I^{(m)}\big(e; E(x_e^{(m)}) \mid E(q)\big) = I^{(m)}\big(e; E(x_e^{(m)})\big) + I^{(m)}\big(e; E(q) \mid E(x_e^{(m)})\big) - I^{(m)}\big(e; E(q)\big). \tag{29}$$

According to Assumption 2 and $I^{(m)}(e; E(q)) \geq 0$, we have:

$$I^{(m)}\big(e; E(x_e^{(m)})\big) \geq I^{(m)}\big(e; E(x_e^{(m)}) \mid E(q)\big) - \delta_{q,e}^{(m)}. \tag{30}$$

Since the mutual information $I^{(m)}\big(e; E(q)\big)$ is only about the view of $\{x_i^{(m)}\}$, we write it as $I\big(e; E(q)\big)$. Combining with Eq. 23, Eq. 24 and Eq. 27 into Eq. 21, we obtain:

$$\begin{aligned}
I^{(m)}(x_e^{(m)}; E(x_e^{(m)})) &\geq I^{(m)}\big(e; E(x_e^{(m)})\big) \\
&\geq I^{(m)}\big(e; E(q), E(\{x_i^{(m)}\})\big) - I(e; E(q)) - \delta_{q,e}^{(m)} \\
&\geq I^{(m)}\big(e; F(\cdot)\big) - I(e; E(q)) - \delta_{q,e}^{(m)} \\
&= H_m - H_m(e|F(\cdot)) - I(e; E(q)) - \delta_{q,e}^{(m)} \\
&\geq H_m - h_b(p_m) - p_m \log_2 k - (1 - p_m)H_{rem} - I(e; E(q)) - \delta_{q,e}^{(m)}
\end{aligned} \tag{31}$$

When the retrieved top-$k$ set does not contain $e$, the remaining candidate space is still large and diverse, and the conditional distribution over entities remains close to the prior. As a result, the residual entropy $H_{rem}$ is well approximated by the prior entropy $H_m$ ($H_{rem} \approx H_m$). Since the intrinsic entropy $H_m$ typically dominates $\log_2 k$ in large-scale knowledge bases, the denominator $H_m - \log_2 k$ is positive and reflects the difficulty of narrowing down the target entity to a small candidate set. Then we can derive an implicit upper constraint on the achievable recall $p_m$:

$$p_m \leq \frac{I^{(m)}(x_e^{(m)}; E(x_e^{(m)})) + I(e; E(q)) + h_b(p_m) + \delta_{q,e}^{(m)}}{H_m - \log_2 k} \leq \frac{I^{(m)}(x_e^{(m)}; E(x_e^{(m)})) + I(e; E(q)) + 1 + \delta_{q,e}^{(m)}}{H_m - \log_2 k}, \quad (32)$$

which is limited by the amount of information $I^{(m)}(x_e^{(m)}; E(x_e^{(m)}))$ and the intrinsic entropy $H_m$ of the view-specific passage representation. Note that $I^{(m)}(e; E(q))$ depends only on the query encoder and input modality, which are shared across different index views. Therefore, when comparing different views $m$, this term can be treated as a task-dependent constant. Intuitively, the bound reveals three key factors that govern the effectiveness of coarse retrieval.

- A better multimodal encoder increases $I^{(m)}(x_e^{(m)}; E(x_e^{(m)}))$, which directly lifts the upper limit of recall for the corresponding view.

- Modalities with higher intrinsic entropy $H_m$ are fundamentally harder to retrieve from, as the target entity remains ambiguous even before retrieval.

- A stronger query encoder, reflected by a larger $I(e; E(q))$, benefits all views uniformly by providing more discriminative signals at the query side.

These factors characterize the trade-off between representation fidelity, modality ambiguity, and query understanding in determining the limits of single-view coarse retrieval.

## A.2. Multi-view upper bound

To analyze the performance of a multi-view indexing system, we consider an "oracle" multi-view coarse retriever that jointly accesses all views and outputs a top-$k$ candidate set, which we denote as $F(E(q), \{\{E(x_i^{(j)})\}_{i=1}^N\}_{j=1}^M, k)$. Treating the oracle retrieval output as a set-decoding variable, Fano's inequality yields an upper bound on the conditional entropy of the ground-truth entity $e$ given the retrieval result:

$$H(e|F(E(q), \{\{E(x_i^{(j)})\}_{i=1}^N\}_{j=1}^M, k)) \leq h_b(p_{ora}) + p_{ora} \log_2 k + (1 - p_{ora})H_{rem}, \quad (33)$$

where $p_{ora}$ denotes the recall@$k$ achieved by the oracle multi-view retriever. For simplicity, we use $F_E(\cdot)$ to denote $F(E(q), \{\{E(x_i^{(j)})\}_{i=1}^N\}_{j=1}^M, k)$. By the definition of mutual information, this bound can be equivalently written as

$$I(e; F_E(\cdot)) = H(e) - H(e|F_E(\cdot)) \quad (34)$$

For simplicity, we write $H := H(e)$ to denote the prior entropy of the target entity. Since the oracle retrieval output $F_E(\cdot)$ is a deterministic function of the query and multi-view knowledge representations, the data processing inequality implies

$$I(e; F_E(\cdot)) \leq I(e; E(q), \{\{E(x_i^{(j)})\}_{i=1}^N\}_{j=1}^M). \quad (35)$$

Applying the chain rule of mutual information, we decompose the available information as

$$I(e; E(q), \{\{E(x_i^{(j)})\}_{i=1}^N\}_{j=1}^M) = I(e; E(q)) + I(e; \{\{E(x_i^{(j)})\}_{i=1}^N\}_{j=1}^M | E(q)), \quad (36)$$

Here we extend assumption 1 to the multi-view level:

$$\begin{aligned} I(e; E(q), \{\{E(x_i^{(j)})\}_{i=1}^N\}_{j=1}^M) &= I(e; E(q)) + I(e; \{\{E(x_i^{(j)})\}_{i=1}^N\}_{j=1}^M | E(q)), \\ &= I(e; E(q)) + I(e; \{E(x_e^{(j)})\}_{j=1}^M | E(q)) + I(e; \{E(\{x_i^{(j)}\}_{i \neq e})\}_{j=1}^M | E(q), \{E(x_e^{(j)})\}_{j=1}^M) \\ &= I(e; E(q)) + I(e; \{E(x_e^{(j)})\}_{j=1}^M | E(q)) \end{aligned}$$

$$(37)$$

We apply the chain rule on the multi-view mutual information:

$$I(e; \{E(x_e^{(j)})\}_{j=1}^M \mid E(q)) = I(e; \{E(x_e^{(j)})\}_{j=1}^M) + I(e; E(q) \mid \{E(x_e^{(j)})\}_{j=1}^M) - I(e; E(q)) \tag{38}$$

Then we extend assumption 2 to the multi-view setting. In particular, when conditioning on the set of target entity representations from all views $\{E(x_e^{(j)})\}_j$, the additional identity information provided by the query may depend on multiple views.

To derive a conservative upper bound, we adopt a worst-case aggregation by allowing the query-induced identity gain to accumulate across views:

$$I(e; E(q) \mid \{E(x_e^{(j)})\}_{j=1}^M) \leq \sum_{j=1}^M \delta_{q,e}^{(j)} \tag{39}$$

This corresponds to an idealized scenario where the identity-related information contributed by the query with respect to different views is maximally non-overlapping. In practice, different views are often correlated and exhibit substantial redundancy, and thus the actual query-induced gain is expected to be significantly smaller than this loose upper bound.

Combining the above equation into Equation 38, we have:

$$I(e; \{E(x_e^{(j)})\}_{j=1}^M) \geq I(e; \{E(x_e^{(j)})\}_{j=1}^M \mid E(q)) + I(e; E(q)) - \sum_{j=1}^M \delta_{q,e}^{(j)} \tag{40}$$

The joint mutual information can be lossely upper bounded by the sum of per-view contributions:

$$I(e; E(x_e^{(1)}), ..., E(x_e^{(M)}) | E(q)) \leq \sum_{j=1}^M I(e; E(x_e^{(j)}) | E(q)) \tag{41}$$

This bound follows from the chain rule and non-negativity of mutual information, and becomes tight only under an idealized setting where modality-specific representations provide conditionally independent and non-overlapping information. In real-world multi-modal retrieval systems, this ideal bound is loose due to two fundamental factors: (i) entropy imbalance across modalities, and (ii) cross-modal redundancy induced by shared semantics. Following (Chen, 2025), entropy imbalance is characterized by $\Delta H$ and modulated by the modality-skew coefficient $\kappa$, which quantifies the penalty introduced by unbalanced modality entropy:

$$\Delta H = \sum_{m=1}^M \big|H_m - H_{\text{bal}}\big|, \tag{42}$$

$$\kappa = \frac{1 - \rho}{M - 1}, \tag{43}$$

where $H_{\text{bal}} = \frac{1}{M} \sum_{m=1}^M H_m$ denotes the entropy under an ideal balanced condition, and $\rho \in [0, 1]$ is the cross-modal redundancy ratio. In particular, $\rho = 0$ (and thus $\kappa = \frac{1}{M-1}$) corresponds to conditionally independent and complementary views, while $\rho = 1$ (and $\kappa = 0$) corresponds to fully redundant modalities.

However, entropy imbalance alone does not capture the reduction of effective information caused by cross-modal redundancy. To account for this effect, we introduce the effective number of modalities $M_{eff}$, which reflects the redundancy-discounted number of independent semantic channels:

$$M_{eff} = 1 + (M - 1)(1 - \rho) \tag{44}$$

The concept of $M_{eff}$ originates from Genome-Wide Association Studies (Li & Ji, 2005). When performing multiple hypothesis tests on Single Nucleotide Polymorphisms (SNPs), SNPs are often highly correlated due to Linkage Disequilibrium. To bridge the gap between the optimistic additive bound and practical redundancy, we adopt an effective-modality approximation and obtain the following redundancy-discounted upper bound:

$$I(e; E(x_e^{(1)}), \ldots, E(x_e^{(M)}) \mid E(q)) \leq \frac{M_{eff}}{M} \sum_{j=1}^M I(e; E(x_e^{(j)}) \mid E(q)) - \kappa \Delta H. \tag{45}$$

Combining the above inequality with Eq. 35 and the chain-rule decomposition, we obtain

$$I\big(e; F_E(\cdot)\big) \leq I\big(e; E(q)\big) + \sum_{j=1}^{M} \delta_{q,e}^{(j)} + \frac{M_{eff}}{M} \sum_{j=1}^{M} +I\big(e; E(x_e^{(j)}) \mid E(q)\big) - \kappa\Delta H. \tag{46}$$

Since $e \to x^{(m)} \to E(x^{(m)})$ forms a Markov chain, by DPI we have $I(e; E(x^{(m)})) \leq I(x^{(m)}; E(x^{(m)}))$. In addition, we treat $I(x^{(m)}; E(x^{(m)}))$ as a conservative upper bound on the per-view contribution, noting that conditioning on the query does not introduce new information about $e$ beyond what is encoded in $x^{(m)}$. For simplicity, we denote $I^{(m)}\big(x^{(m)}; E(x^{(m)})\big)$ as $I^{(m)}(\cdot)$. Substituting into the above bound yields

$$I\big(e; F_E(\cdot)\big) \leq I\big(q; E(q)\big) + \sum_{j=1}^{M} \delta_{q,e}^{(j)} + \frac{M_{eff}}{M} \sum_{j=1}^{M} I^{(j)}(\cdot) - \kappa\Delta H. \tag{47}$$

Combining Eq. 33 and Eq. 34 into the above equation gives

$$I\big(e; F_E(\cdot)\big) = H(e) - H\big(e \mid F_E(\cdot)\big) \geq H - h_b(p_{\text{ora}}) - p_{\text{ora}} \log_2 k - (1 - p_{\text{ora}})H_{\text{rem}}. \tag{48}$$

Combining Eq. 47 and Eq. 48 yields an implicit constraint on $p_{\text{ora}}$:

$$H - h_b(p_{\text{ora}}) - p_{\text{ora}} \log_2 k - (1 - p_{\text{ora}})H_{\text{rem}} \leq I\big(e; E(q)\big) + \sum_{j=1}^{M} \delta_{q,e}^{(j)} + \frac{M_{eff}}{M} \sum_{j=1}^{M} I^{(j)}(\cdot) - \kappa\Delta H. \tag{49}$$

For large-scale knowledge bases, when the retrieved top-$k$ set does not contain $e$, the remaining candidate space is still large and diverse, and the residual entropy is well approximated by the prior, i.e., $H_{\text{rem}} \approx H$. Then the above inequality simplifies to

$$p_{\text{ora}}\big(H - \log_2 k\big) \leq \frac{M_{eff}}{M} \sum_{j=1}^{M} I^{(j)}(\cdot) + \sum_{j=1}^{M} \delta_{q,e}^{(j)} + I\big(e; E(q)\big) - \kappa\Delta H + h_b(p_{\text{ora}}). \tag{50}$$

Using $h_b(p_{\text{ora}}) \leq 1$, we obtain a concise upper bound:

$$p_{\text{ora}} \leq \frac{\frac{M_{eff}}{M} \sum_{j=1}^{M} I^{(j)}(\cdot) + \sum_{j=1}^{M} \delta_{q,e}^{(j)} + I\big(e; E(q)\big) - \kappa\Delta H + 1}{H - \log_2 k}. \tag{51}$$

Eq. 51 characterizes the theoretical upper limit of multi-view coarse retrieval. The essential difference between single-view and multi-view retrieval lies in how view-specific index information is aggregated. The first is the redundancy-discounted aggregation term $\frac{M_{eff}}{M} \sum_{j=1}^{M} I^{(j)}(\cdot)$, where $M_{\text{eff}}$ represents the effective number of independent semantic channels. This term captures how complementary information from multiple views can increase the retrieval ceiling beyond any single index. In the ideal case of conditionally independent views, $M_{\text{eff}} \to M$ and the bound approaches the fully additive regime. In contrast, when views are highly redundant, $M_{\text{eff}} \to 1$, and the benefit of multi-view indexing collapses to that of a single view.

The imbalance term appears as a penalty $\kappa\Delta H$, but its interaction with redundancy is subtle. As redundancy increases ($\rho \uparrow$), the skew coefficient decreases ($\kappa = \frac{1-\rho}{M-1} \downarrow$), which attenuates the imbalance penalty. This is consistent with the intuition that highly overlapping views tend to exhibit less effective imbalance in practice. However, the same redundancy simultaneously reduces the effective aggregation gain through $M_{\text{eff}} = 1 + (M-1)(1-\rho)$, shrinking the factor $\frac{M_{\text{eff}}}{M}$ from 1 (independent views) toward $\frac{1}{M}$ (fully redundant views). Therefore, while redundancy may lessen the imbalance penalty, it more fundamentally limits multi-view benefits by collapsing the number of effective independent semantic channels, which dominates the overall reduction of the recall ceiling.

Together, this analysis shows that multi-view retrieval improves the theoretical upper bound only when different views contribute complementary and sufficiently balanced information. Otherwise, the system degenerates gracefully to the single-view regime. This observation explains both the potential gains and the inherent limitations of multi-index coarse retrieval, and motivates the need for mechanisms that adaptively exploit cross-view complementarity.

**System:**

**[E-VQA]** Answer the encyclopedic question about the given image. Don't mention the visual content of image in your output. Directly output the answer of the question according to the Context. If the context does not contain the information required to answer the question, you should answer the question using internal model knowledge.

**[InfoSeek]** Answer the encyclopedic question about the given image. Don't mention the visual content of the image in your output. Directly output the answer of the question according to the context.

You should answer the question using internal model knowledge.

If you need to answer questions about numbers or time, please output the corresponding numerical format directly. If the context does not contain the information required to answer the question, you should answer the question using internal model knowledge.

There is an example:

- Context: # Wiki Article: Dolomites

## Section Title: Dolomites

The Dolomites, also known as the Dolomite Mountains, Dolomite Alps or Dolomitic Alps, are a mountain range located in northeastern Italy. The Dolomites are located in the regions of Veneto, Trentino-Alto Adige/Südtirol and Friuli Venezia Giulia, covering an area shared between the provinces of Belluno, Vicenza, Verona, Trentino, South Tyrol, Udine and Pordenone.

- Question: Which city or region does this mountain locate in?

Just answer the questions , no explanations needed. Short answer is: Province of Belluno

---

**User:**

*Query Image*

-Context: *Entity Section*

-Question: *Textual Question*

**[E-VQA]** The answer is:

**[InfoSeek]** Just answer the questions , no explanations needed. Short answer is:

*Table 7.* Prompt used for visual question answering. Blue part is the instructions for E-VQA, green part is the instructions for InfoSeek.

## B. Implementation Details

### B.1. Datasets

**E-VQA (Mensink et al., 2023)** encompasses 221k visual question-answer pairsa bout more than 10k entities which is derived from the iNaturalist2021 and the Goolge Landmarks Dataset V2 (Van Horn et al., 2021; Weyand et al., 2020). (Mensink et al., 2023) also provides a knowledge base with 2M Wikipedia articles with images. To be consistent with related works, We use 4750 single-hop questions for testing. We report Recall@K to evaluate the retrieval performance. And use the BEM score that provided by E-VQA dataset evaluating QA performance.

**InfoSeek (Chen et al., 2023)** comprises 1.3 million QA pairs, which is derived from OVEN (Hu et al., 2023). Due to the lack of ground truth for test split, our experiments evaluation is conducted on the validation set which contains 71,355 questions. In particular, both the validation and test sets feature questions pertaining to unseen entities or queries. To be consisting with related works (Yan & Xie, 2024; Yang et al., 2025b), we use the subset of 100k entities as the knowledge base during retrieval. For VQA performance, we report the VQA accuracy metric (Antol et al., 2015), which report whether MLLM's answer exactly matches any of valid answers. And for questions about numeric, we follow (Yang et al., 2025b) to use the relaxed accuracy (Methani et al., 2020), which considers an answer correct if it falls within the ground truth range.

### B.2. QA prompts

In this section, we present the prompt used in MLLM answer generation. Follow (Yang et al., 2025b) settings, the entire template is structured in an agentlike format. During the question-answering phase, we use exactly the same prompting strategy across all retrieval configurations to ensure a fair comparsion. This design guarantees that any observed improvement in QA performance is solely attributable to differences in the retrieval stage, rather than variations in prompt design or

*Table 8.* Experiment result with re-ranking module on E-VQA and InfoSeek.

| Method | E-VQA | | | | InfoSeek | | | |
|---|---|---|---|---|---|---|---|---|
| | R@1 | R@5 | R@10 | R@20 | R@1 | R@5 | R@10 | R@20 |
| Title View | 29.6 | 38.9 | 40.0 | 44.7 | 60.3 | 73.9 | 76.0 | 78.9 |
| Image View | 32.6 | 45.5 | 46.8 | 48.9 | 56.4 | 72.5 | 75.1 | 76.0 |
| EchoSight | 36.5 | 47.9 | 48.8 | 48.8 | 53.2 | 74.0 | 77.4 | 77.9 |
| OMGM$^\diamond$ | 39.0 | 51.5 | 53.7 | 58.7 | **62.6** | 78.5 | 81.2 | 84.7 |
| Ours | **43.1** | **58.1** | **59.7** | **65.1** | 62.5 | **81.4** | **85.8** | **87.7** |

*Table 9.* Experimental results of different views combinations on E-VQA and InfoSeek

| Image | Title | Summary | E-VQA | | | | InfoSeek | | | |
|---|---|---|---|---|---|---|---|---|---|---|
| | | | R@1 | R@5 | R@10 | R@20 | R@1 | R@5 | R@10 | R@20 |
| ✓ | | | 13.4 | 31.8 | 41.8 | 48.8 | 45.7 | 68.6 | 74.6 | 76.0 |
| | ✓ | | 17.6 | 31.9 | 38.7 | 44.7 | 51.3 | 69.2 | 74.8 | 78.9 |
| | | ✓ | 18.7 | 41.2 | 49.7 | 58.7 | 52.2 | 73.7 | 79.8 | 84.7 |
| ✓ | ✓ | | 25.4 | 44.3 | 52.0 | 59.4 | 57.6 | 76.7 | 82.0 | 85.5 |
| ✓ | | ✓ | **28.8** | 46.5 | 54.4 | 62.9 | 58.5 | 77.0 | 82.7 | 86.3 |
| | ✓ | ✓ | 22.3 | 38.0 | 46.3 | 54.9 | 55.2 | 73.5 | 79.8 | 84.4 |
| ✓ | ✓ | ✓ | **28.8** | **49.9** | **57.0** | **65.1** | **60.8** | **80.0** | **84.6** | **87.7** |

answer generation.

## B.3. Implementation Details

To be consistent with previous works (Yang et al., 2025b; Yan & Xie, 2024; Cocchi et al., 2025), we use pre-trained EVA-CLIP-8B (Sun et al., 2024) encoder to build embeddings for query images and entity titles, images and summaries. In the additional re-ranking stage, we utilize open-source OMGM re-ranking module as the multi-modal re-ranking (Yang et al., 2025b). In the generation stage, we choose to use only a section of the top-1 entity as the context to answer the question. Specifically, we follow OMGM step 3, using a pre-trained BGE-Reranker-v2-m3 to extract the most relevant section in the top-1 entity. The BGE model is a lightweight and efficient text re-ranking model, which is frozen in our pipeline.

Regarding the MLLM answer generator, we test multiple MLLM backbones, including common used LLaVA (Liu et al., 2024) and most advanced backbones Qwen3-VL 8B (Yang et al., 2025a) and InterVL-3.5 8B (Wang et al., 2025). All MLLMs used in our experiments are obtained from their official releases on HuggingFace, and no additional finetuning or task-specific training is performed at any stage of the pipeline. Some prior work in KB-VQA evaluates generalization by testing text-only large language models, where the answer generation stage receives only textual inputs together with retrieved entity sections. While such settings are simpler to evaluate and easier to reproduce, we argue that perceiving and understanding visual content is an indispensable component of KB-VQA, as the task fundamentally requires grounding questions in images. Therefore, instead of adopting a text-only evaluation, we select three representative MLLMs as answer generators to provide a more faithful and rigorous assessment.

## C. Additional Experimental Results

**Re-ranking on E-VQA and InfoSeek.** Here we report the experimental results with re-ranking module on E-VQA and InfoSeek in Table 8. Also, we report the results of different views combination in Table 9. As shown in Table 8 and Table 9, SCAR consistently outperforms single-view retrieval baselines across all evaluated datasets. The performance trends are consistent with those observed in the main experiments, indicating that the proposed method generalizes well to different multimodal knowledge bases.

**VQA performance on E-VQA and InfoSeek.** Table 10 compares the KB-VQA performance of our SCAR-based KB-VQA pipeline with state-of-the-art methods on E-VQA and InfoSeek. As shown in the table, the majority of existing approaches rely on finetuning the answer generator, introducing task-specific preprocessing, or incorporating reranking modules that are trained on the target benchmarks. To ensure a transparent and fair comparison, we explicitly annotate whether each method involves generator finetuning (Gen. FT) or retrieval-related finetuning (Ret. FT).

In this work, we adopt a strict definition of retrieval finetuning: any module before the MLLM receives retrieved context is considered finetuned if it was trained on E-VQA or InfoSeek datasets. Under this definition, methods utilizing retrievers, preprocessors, or re-rankers trained on the target datasets are categorized as retrieval-finetuned.

In SCAR, the multi-view retrieval operates in a fully finetuning-free manner. Our re-ranking variants optionally employ the frozen OMGM (Yang et al., 2025b) reranking module using the officially released checkpoint, without any additional training or task adaptation. We treat these variants as retrieval finetuned, and clearly distinguish them from the fully training-free setting.

The results in Table 10 demonstrate that SCAR significantly strengthens the KB-VQA pipeline and achieves performance comparable to, or even surpassing, finetuned baselines without requiring any training. This highlights the critical role of improved coarse-grained retrieval in KB-VQA. Since the answer generator or any re-ranking module cannot recover

*Table 10.* Comparison with state-of-the-art KB-VQA methods on E-VQA and InfoSeek. Rows highlighted in blue indicate the best fully training-free baseline (no finetuning on generator or retriever), while rows highlighted in green indicate the best generator training-free baseline. **Bold** denotes the overall best performance and underline denotes the overall second best performance. *w. reranking* denotes using a frozen OMGM (Yang et al., 2025b) re-ranking module without further training.

| Method | Generator Model | Retriever | Retrieval Mode | Gen. FT | Ret. FT | E-VQA | Infoseek Unseen-Q | Infoseek Unseen-E | Overall |
|---|---|---|---|---|---|---|---|---|---|
| RoRA-VLM (Qi et al., 2024) | LLaVA-1.5-7B | CLIP+Google Search | Visual+Textual | ✓ | ✗ | 20.3 | 27.3 | 25.1 | - |
| DPR$_{V+T}$ (Lerner et al., 2024) | Mutli-passage Bert | CLIP ViT-B/32 | Visual+Textual | ✓ | ✓ | 29.1 | - | - | 12.4 |
| Wiki-LLaVA (Caffagni et al., 2024) | LLaVA-1.5-7B | CLIP ViT-L/14+Contriever | Textual | ✓ | ✗ | 21.8 | 30.1 | 27.8 | 28.9 |
| EchoSight (Yan & Xie, 2024) | LLaVA-1.5-7B | EVA-CLIP-8B | Visual | ✗ | ✓ | 24.9 | 18.0 | 19.8 | 18.8 |
| mR$^2$AG (Zhang et al., 2024) | LLaVA-1.5-7B | CLIP-ViT-L/14@336px | Visual+Textual | ✓ | ✓ | - | 40.6 | 39.8 | 40.2 |
| ReflectiVA (Cocchi et al., 2025) | LLaVA-MORE-7B | EVA-CLIP-8B | Visual/Textual | ✓ | ✓ | 35.5 | 40.4 | 39.8 | 40.1 |
| OMGM (Yang et al., 2025b) | LLaVA-1.5-7B | EVA-CLIP-8B | Textual(Summary) | ✓ | ✓ | **50.2** | **43.5** | **43.5** | **43.5** |
| OMGM (Yang et al., 2025b) | LLaVA-1.5-7B | EVA-CLIP-8B | Textual(Summary) | ✗ | ✓ | 49.9 | 35.3 | 33.6 | 34.4 |
| mKG-RAG (Yuan et al., 2025) | LLaVA-MORE-7B | QM-Retriever | Visual+Textual | ✓ | ✓ | 36.3 | 41.4 | 39.6 | 40.5 |
| MMKB-RAG (Ling et al., 2025) | Qwen-2-7B | EVA-CLIP-8B | Visual | ✓ | ✗ | 35.9 | 36.4 | 36.3 | 36.4 |
| Wiki-PRF (Hong et al., 2025) | Qwen-2.5VL-7B | EVA-CLIP-8B | Visual | ✓ | ✓ | 28.6 | 40.0 | 39.4 | 39.5 |
| VLM-PRF (Hong et al., 2025) | InternVL3-8B | EVA-CLIP-8B | Visual | ✓ | ✓ | 39.2 | 43.3 | 42.1 | 42.5 |
| SCAR(ours) | LLaVA-1.5-7B | EVA-CLIP-8B | Mutli-view | ✗ | ✗ | 34.4 | 31.5 | 30.5 | 31.0 |
| SCAR(ours) | *w. reranking* | EVA-CLIP-8B | Mutli-view | ✗ | ✓ | 42.8 | 34.5 | 32.0 | 33.2 |
| SCAR(ours) | Qwen3-VL-8B | EVA-CLIP-8B | Mutli-view | ✗ | ✗ | 39.8 | 34.1 | 34.9 | 34.5 |
| SCAR(ours) | *w. reranking* | EVA-CLIP-8B | Mutli-view | ✗ | ✓ | 49.3 | 38.3 | 36.4 | 37.3 |
| SCAR(ours) | InterVL-3.5-8B | EVA-CLIP-8B | Mutli-view | ✗ | ✗ | 39.0 | 31.9 | 32.6 | 32.3 |
| SCAR(ours) | *w. reranking* | EVA-CLIP-8B | Mutli-view | ✗ | ✓ | 50.1 | 35.3 | 33.9 | 34.6 |

from missing ground truth entities in the retrieved candidates, retrieval quality fundamentally bounds the achievable QA performance.

The improvement brought by multi-view retrieval is particularly pronounced on E-VQA, where entity ambiguity and diverse visual–textual cues make single-view retrieval insufficient. On InfoSeek, where the knowledge base we used contains only around 100K entities, retrieval performance across methods tends to saturate. Moreover, InfoSeek adopts a strict exact-match evaluation protocol (Cocchi et al., 2025), which amplifies the benefit of MLLM generator finetuning. As a result, finetuned methods such as OMGM show substantial gains from generator finetuning on InfoSeek, with an improvement of up to 9% compared to their non-finetuned counterparts.

Overall, these observations confirm that enhancing retrieval coverage through multi-view indexing is a decisive factor for KB-VQA performance, regardless of whether reranking modules are incorporated or the MLLM generator is finetuned.

### C.1. Additional Ablation Study

As Figure 3 shows, single-view baselines also improve as the candidate pool expands, and the Summary view consistently outperforming the others. This observation is consistent with the single-view upper bound analysis in Eq. 5, suggesting that single-modality retrieval performance is constrained by the amount of mutual information captured by the encoder in the corresponding view.

The table 13 reports the frequency with which redundancy regulation is triggered between different pairs of views during retrieval. Among all view pairs, the Summary–Title combination exhibits the highest trigger count, indicating stronger structural similarity between their induced entity kNN graphs. This observation aligns with the intuition that text-based views, sharing similar semantic sources, are more prone to structural redundancy. In contrast, view pairs involving the Image view trigger redundancy regulation less frequently, suggesting that visual information provides more complementary structural signals. These results further support our priority analysis, highlighting that separating structurally redundant text views can effectively mitigate redundant competition and improve overall retrieval performance.

We additionally measure the runtime overhead of SCAR as the retrieval scope k increases, recording the average per-query computation time under different candidate sizes. Empirically, the runtime of SCAR grows approximately quadratically with respect to k, i.e., O($k^2$), due to the construction and processing of view-specific kNN graphs and cross-view structural similarity computation. Specifically, the average per-query runtime increases from 6.40 ms at k=20 to 234 ms at k=200. However, we observe that when k exceeds approximately 50, the improvement in retrieval upper bounds becomes marginal. In this regime, retrieval performance is primarily limited by the representational capacity of the underlying encoders rather than the candidate pool size, and further enlarging k yields diminishing returns. Importantly, even at k=200, the additional computation introduced by SCAR remains a minor component of the overall KB-VQA pipeline, accounting for only a small

*Table 11.* Retrieval performance on WebQA and OVEN measured by Recall@5 under the M-BEIR setting. Results show that combining text and image views yields superior performance compared to single-view retrieval.

| Text | Image | WebQA $text \rightarrow (image, text)$ | OVEN $image \rightarrow (image, text)$ |
|:---:|:---:|:---:|:---:|
| ✓ | | 49.3 | 60.0 |
| | ✓ | 43.7 | 52.2 |
| ✓ | ✓ | **56.0** | **63.2** |

*Table 12.* Ablation study experimental results on E-VQA and InfoSeek

| Module I | Module II | E-VQA | | | | InfoSeek | | | |
|:---:|:---:|:---:|:---:|:---:|:---:|:---:|:---:|:---:|:---:|
| | | R@1 | R@5 | R@10 | R@20 | R@1 | R@5 | R@10 | R@20 |
| | | 24.6 | 42.3 | 46.4 | 56.4 | 54.7 | 74.8 | 77.3 | 84.6 |
| ✓ | | 28.4 | 48.3 | 56.3 | 63.8 | 60.0 | 79.4 | 84.3 | 87.5 |
| ✓ | ✓ | **28.8** | **49.9** | **57.0** | **65.1** | **60.8** | **80.0** | **84.6** | **87.7** |

*Table 13.* Frequency of redundancy regulation triggers between views on E-VQA.

| View Pair | Trigger Count |
|:---:|:---:|
| Summary - Image | 3640 |
| Summary - Title | 4412 |
| Image - Title | 3217 |

fraction of the total runtime compared to downstream reranking and answer generation stages. This indicates that SCAR provides a favorable trade-off between retrieval quality and computational cost, while remaining practical for inference-time deployment.

**Generalization on Additional Benchmark.** Many multimodal retrieval tasks also involve a multimodal knowledge base. We therefore conduct additional experiments on WebQA (Chang et al., 2022) and OVEN (Hu et al., 2023) to evaluate its generalization ability. All experiments follow the M-BEIR (Wei et al., 2024) settings, and the same hyperparameters as those used in the main experiments on E-VQA and InfoSeek are adopted without further tuning. For each dataset, we construct separate text and image views and apply SCAR accordingly. As shown in Table 11, SCAR with two-view integration consistently outperforms the single-view baselines, demonstrating that our method effectively adapts to multimodal knowledge bases and yields tangible improvements in retrieval performance.

## C.2. Additional Hyperparameter Sensitivity Analysis.

In this appendix, we provide additional hyperparameter sensitivity results to complement the analysis in the main paper.

**kNN graph size $n$.** We vary the number of neighbors n from 3 to 20 while keeping all other hyperparameters fixed. As n increases from small values, retrieval performance improves consistently, indicating that overly sparse graphs limit effective neighborhood propagation. Performance peaks around a moderate value ($k \approx 10$) and then saturates, with only minor fluctuations for larger n. Importantly, we observe no increase in runtime as n grows, suggesting that the proposed method is computationally stable with respect to the graph size. Based on this trade-off, we fix $n = 10$ in all experiments. Full results are reported in Table 16.

**Full hyperparameter analysis.** While key trends are visualized in the main text, we report full numerical results here for completeness and reproducibility. Across all settings, SCAR exhibits consistent performance within reasonable parameter ranges, further supporting its robustness and the conclusions drawn in the main paper. Specifically, Table 14 reports the detailed results under different diffusion strengths. In addition to the hyperparameters discussed in the main text, we further report sensitivity analyses for the view-level reduction factor $\gamma$, the graph similarity threshold $\tau$, and the entity-level suppression factor $\beta$ in Tables 15, 17, and 18, respectively.

## C.3. Case Study on Multi-view Retrieval Complementarity

We present more qualitative case studies to illustrate how SCAR improves coarse-grained retrieval by integrating complementary signals from different view-specific indexes. Figure 6 visualizes queries together with the top retrieved entities from image, title, and summary based indexes, as well as the final result produced by SCAR.

Across all cases, we observe that SCAR effectively aggregates informative views while suppressing noisy or redundant signals, leading to more reliable retrieval outcomes.

For the query "What is the habitat of this bird?", the image-based index retrieves visually similar but semantically irrelevant entities, while the title-based index suffers from lexical ambiguity, retrieving a related but incorrect species. The summary-

| Query | Retrieval Result | | | |
|---|---|---|---|---|
| | Image index | Title index | Summary index | SCAR (ours) |
| What is the habitat of this bird? | ☒ List of Ciconiiformes by population | ☒ Bare-faced ibis | ☑ Buff-necked ibis | ☑ Buff-necked ibis |
| What is this plant's relationship to central europe? | ☒ Sodenberg | ☑ Leucojum vernum | ☒ Soldanella | ☑ Leucojum vernum |
| Besides deer, what browses the leaves of this plant? | ☑ Balsamorhiza sagittata | ☒ Arnica longifolia | ☒ Wyethia mollis | ☑ Balsamorhiza sagittata |

*Figure 6.* Case studies of multi-view retrieval for KB-VQA. We visualize representative queries and the corresponding top retrieved entities from image, title, and summary based indexes, together with SCAR's final retrieval. ✓ / ✗ denote correct and incorrect retrievals, respectively. While each single-view index captures partial and complementary semantic cues, none is consistently reliable. By coordinating multi-view evidence, SCAR effectively aggregates the strengths of multi indexes and mitigates their failure cases.

based index successfully identifies the correct entity (Buff-necked ibis) by capturing high-level semantic descriptions related to habitat. SCAR preserves this informative signal and filters out conflicting candidates from other views.

In the query concerning the plant's relationship to Central Europe, the title-based index correctly retrieves "Leucojum vernum" due to precise taxonomic naming, whereas the image and summary based indexes retrieve visually or descriptively similar but geographically incorrect species. SCAR successfully retains the discriminative signal from the title view while preventing interference from visually similar but semantically misleading candidates.

For the query "Besides deer, what browses the leaves of this plant?", the image-based index correctly retrieves "Balsamorhiza sagittata" by recognizing distinctive visual features, while the title and summary views retrieve related but incorrect plant species. SCAR identifies the image view as the most informative in this case and recovers the correct entity.

*Table 14.* Sensitivity of SCAR to the diffusion strength $\alpha$.

| Diffusion Strength $\alpha$ | R@1 | R@5 | R@10 | R@20 |
|---|---|---|---|---|
| 0 | 25.7 | 47.3 | 55.8 | 63.1 |
| 0.10 | 25.7 | 48.0 | 56.0 | 63.2 |
| 0.15 | 25.7 | 47.8 | 56.1 | 63.3 |
| 0.20 | 25.6 | 48.0 | 56.1 | 62.9 |
| 0.25 | 26.1 | 48.2 | 56.2 | 62.9 |
| 0.30 | 25.7 | 48.7 | 56.3 | 63.0 |
| 0.35 | 28.9 | 48.6 | 56.4 | 62.9 |
| 0.40 | 26.4 | 48.9 | 56.5 | 62.9 |
| 0.45 | 26.6 | 49.3 | 56.7 | 62.9 |
| 0.50 | 26.8 | 49.3 | 56.8 | 63.1 |
| 0.55 | 26.9 | 49.2 | 57.1 | 63.3 |
| 0.60 | 27.8 | 49.3 | 57.0 | 63.3 |
| 0.65 | 27.9 | 49.8 | 57.3 | 63.4 |
| 0.70 | 28.1 | 49.8 | 57.2 | 64.1 |
| 0.75 | 28.4 | 50.3 | 57.5 | 64.0 |
| 0.80 | 28.8 | 49.9 | 57.7 | 64.1 |
| 0.85 | 29.6 | 50.0 | 57.2 | 64.4 |
| 0.87 | 28.9 | 50.2 | 57.1 | 64.6 |
| 0.90 *(default)* | 28.9 | 50.0 | 57.0 | 65.0 |
| 0.92 | 28.1 | 49.8 | 57.0 | 64.9 |
| 0.95 | 28.0 | 49.8 | 57.0 | 64.3 |
| 0.97 | 26.3 | 49.0 | 56.9 | 63.9 |

*Table 15.* Sensitivity of SCAR to view-level reduction factor $\gamma$.

| View-level Reduction Factor $\gamma$ | R@1 | R@5 | R@10 | R@20 |
|---|---|---|---|---|
| 0.50 | 28.2 | 49.8 | 56.8 | 62.7 |
| 0.55 | 28.8 | 49.9 | 56.9 | 63.1 |
| 0.60 | 28.8 | 50.0 | 56.8 | 63.2 |
| 0.65 | 28.7 | 49.9 | 57.1 | 64.1 |
| 0.70 | 28.8 | 49.8 | 57.0 | 64.5 |
| 0.75 | 28.8 | 49.9 | 57.0 | 64.5 |
| 0.80 *(default)* | 28.9 | 50.0 | 57.0 | 65.0 |
| 0.85 | 28.9 | 50.0 | 57.0 | 64.7 |
| 0.90 | 28.7 | 50.0 | 56.9 | 64.5 |
| 0.95 | 28.6 | 49.9 | 57.1 | 64.4 |

*Table 16.* Sensitivity of SCAR to the kNN graph size.

| kNN Graph size | R@1 | R@5 | R@10 | R@20 | Running Time/ms |
|---|---|---|---|---|---|
| 3 | 26.3 | 49.4 | 56.5 | 63.8 | 5.26 |
| 5 | 28.4 | 50.0 | 57.1 | 64.1 | 5.22 |
| 7 | 29.2 | 50.0 | 56.9 | 64.3 | 5.36 |
| 10 *(default)* | 28.9 | 50.0 | 57.0 | 65.0 | 5.33 |
| 13 | 28.8 | 50.3 | 57.3 | 64.7 | 5.21 |
| 15 | 29.0 | 50.3 | 57.0 | 64.6 | 5.05 |
| 17 | 29.4 | 50.3 | 57.2 | 64.6 | 4.99 |
| 20 | 29.4 | 50.4 | 56.7 | 64.7 | 4.99 |

*Table 17.* Sensitivity of SCAR to the similarity threshold $\tau$.

| Similarity Threshold $\tau$ | R@1 | R@5 | R@10 | R@20 |
|---|---|---|---|---|
| 0.50 | 28.9 | 50.3 | 57.1 | 65.0 |
| 0.55 | 28.9 | 50.3 | 57.1 | 65.0 |
| 0.60 | 28.9 | 50.3 | 57.1 | 65.0 |
| 0.65 | 29.0 | 50.2 | 57.0 | 65.0 |
| 0.70 *(default)* | 28.9 | 50.0 | 57.9 | 65.0 |
| 0.75 | 28.9 | 50.0 | 57.0 | 64.8 |
| 0.80 | 28.9 | 50.1 | 57.1 | 64.8 |
| 0.85 | 28.6 | 50.1 | 57.2 | 64.8 |
| 0.90 | 29.1 | 48.9 | 57.3 | 64.7 |
| 0.95 | 27.8 | 48.8 | 57.3 | 64.6 |

*Table 18.* Sensitivity of SCAR to entity-level suppression factor $\beta$.

| Entity-level Suppression Factor $\beta$ | R@1 | R@5 | R@10 | R@20 |
|---|---|---|---|---|
| 0.05 | 27.7 | 50.0 | 56.9 | 65.0 |
| 0.10 | 27.9 | 50.1 | 56.9 | 65.0 |
| 0.15 | 28.3 | 50.0 | 56.9 | 65.0 |
| 0.20 *(default)* | 28.9 | 50.0 | 57.0 | 65.0 |
| 0.25 | 29.0 | 49.9 | 56.9 | 65.0 |
| 0.30 | 28.5 | 50.1 | 56.9 | 64.9 |
| 0.35 | 28.6 | 50.3 | 56.9 | 64.9 |
| 0.40 | 28.7 | 50.1 | 57.0 | 64.9 |
| 0.45 | 28.8 | 50.1 | 57.0 | 64.9 |
| 0.50 | 28.8 | 50.1 | 57.1 | 64.9 |
| 0.55 | 28.8 | 50.1 | 57.1 | 64.9 |
| 0.60 | 29.0 | 49.9 | 57.0 | 64.9 |

