# OpenReview forum: "Beyond Single-View Indexing: Structure-Aware Multi-View Retrieval for Knowledge-Based VQA"
_ICML.cc/2026/Conference — ICML 2026 regular_

### Official Review · Reviewer_5yLT · 2026-02-26

**Soundness:** 3
**Presentation:** 3
**Significance:** 3
**Originality:** 3
**Overall Recommendation:** 4
**Confidence:** 4

**Summary:**

This paper addresses the "coverage gap" in KB-VQA retrieval. It proposes SCAR, a training-free framework that integrates Image, Title, and Summary views. By combining Similarity Propagation (refining scores via kNN graphs) and Redundancy Regulation (suppressing overlapping info), SCAR significantly boosts retrieval recall on E-VQA and InfoSeek with minimal overhead.

**Compliance With Llm Reviewing Policy:**

Affirmed.

**Key Questions For Authors:**

1.  Eq. 6 includes an imbalance penalty $\kappa \Delta H$. Could you clarify if SCAR’s regulation module implicitly addresses the "entropy imbalance" mentioned in your theory, or is it primarily focused on the $M_{eff}$ term? Specifically, does the scaling factor $\gamma$ account for the modality-skew coefficient $\kappa$?
2.  In Eq. 10, you exclude entities absent from a view by removing all incident edges. Does this "all-or-nothing" approach prevent potentially correct entities from benefiting from cross-view similarity propagation if they are only retrieved by a single, lower-priority view?
3. You used **EVA-CLIP-8B** for all views. If the Title and Summary views were encoded by a dedicated text-only encoder (e.g., BGE) while the Image view used CLIP, would the structural similarity metric ($sim(j_1, j_2)$) still be reliable given the different embedding spaces?

**Limitations:**

yes

**Strengths And Weaknesses:**

Strengths:

1. The paper provides a formal information-theoretic upper bound for multi-view retrieval. By introducing the concept of the "effective number of independent views", the authors provide a mathematical basis for why multi-view retrieval benefits from non-redundant information.

2.Unlike naive score fusion, SCAR uses manifold diffusion (Eq. 12) to reinforce entities within strong semantic neighborhoods.  This effectively suppresses "hallucinated" entities that lack structural support within a specific view.

3.As a training-free method, SCAR is highly practical for real-world deployments. It adds only ~40ms of latency per query on E-VQA, accounting for less than 0.5% of the total inference time.

Weaknesses

1. The runtime of SCAR grows approximately quadratically ($O(k^2)$) with the candidate size $k$ due to the construction and processing of view-specific kNN graphs. This may pose a bottleneck if a much larger candidate pool is required for extremely complex knowledge bases.

2. The algorithm excludes entities not observed in a given view by removing all incident edges in the kNN graph. This "all-or-nothing" approach might prevent potentially correct entities from benefiting from cross-view similarity propagation if they are missed by a high-priority view.

---

> ### Author Rebuttal · Authors · 2026-03-31
>
> ## Concern about SCAR's quadratic runtime scaling at large k.(W1)
>
> We agree that the runtime of SCAR increases with the candidate size k due to the construction and processing of view-specific kNN graphs, leading to an approximate O(k^2) complexity. To quantify this behavior, **we measured per-query runtime under different candidate sizes (k from 20 to 200).**
>
> |Running Time||
> |-|-|
> |k|Time (millisecond)|
> |20|6.40|
> |30|16.65|
> |40|17.49|
> |50|24.46|
> |60|31.69|
> |80|50.84|
> |100|72.48|
> |120|99.56|
> |150|140.26|
> |180|191.62|
> |200|233.91|
>
> A quadratic fit yields: $t(k)=0.0044k^2+0.27k+0.73$ ($R^2=0.999$), confirming the expected growth trend.
>
> **However, within the tested range up to k=200, the absolute runtime of SCAR remains below 250 ms per query.** In comparison, **downstream reranking and answer generation stages require several seconds (e.g., around 6000–8000 ms per query),** making the additional overhead introduced by SCAR a small fraction of the total latency.
>
> Moreover, retrieval performance shows diminishing returns as k grows. Therefore, extremely large candidate pools are rarely required in practice, and the quadratic complexity does not constitute a practical bottleneck.
>
> ## Concern about all-or-nothing exclusion blocking cross-view recovery.(W2&Q2)
>
> We thank the reviewer for raising this important point. We clarify that **the proposed design does not remove entities from the global candidate pool when they are absent from a particular view.** Each view constructs its kNN graph independently, and missing entities are simply excluded from propagation within that view. This design prevents entities from being influenced by unrelated neighbors and introducing noise during diffusion. **Importantly, entities missed by one view can still participate fully in other views where they are observed.** Cross-view regulation is applied only to entities appearing in both views, so entities absent from a high-priority view are not penalized.
>
>
> Furthermore, **we empirically observe that SCAR is robust to the choice of view priority.** As reported in our paper, evaluating all permutations of the three views results in only minor performance variations. This result further indicates that **no single view dominates the retrieval process and complementary signals can recover missed entities.**
>
> ## Question about how $\kappa\Delta H$, $M_{eff}$, and $\gamma$ map in SCAR.(Q1)
>
> We would like to clarify that **Eq. 6 provides a theoretical decomposition of multi-view retrieval behavior, rather than a direct one-to-one implemnetation for algorithms.** In particular, the term $\kappa \Delta H$ is introduced to maintain theoretical consistency with prior analyses[1] and to characterize the potential impact of modality imbalance at a conceptual level.
>
> In SCAR, **the cross-view regulation module primarily addresses the redundancy aspect captured by the $M_{eff}$ term.** By detecting structural overlap and suppressing redundant views, the method increases the number of independent semantic views, aligning with the role of $M_{eff}$ in the analysis. The parameter $\gamma$ should therefore be interpreted as a parameter for redundancy regulation rather than an explicit implementation of $\kappa$.
>
>
> ## Concern about structural similarity under heterogeneous embedding spaces.(Q3)
>
> We thank the reviewer for raising this question **regarding the reliability of the structural similarity metric under heterogeneous embedding spaces.**
>
> In SCAR, the structural similarity $sim(j_1, j_2)$ is computed over affinity matrices constructed within each view. Therefore, **the comparison is based on relational neighborhood structure rather than absolute embedding coordinates.** As a result, the metric does not require embeddings from a shared feature space. To empirically validate this property, we conducted an additional experiment in which **the Title view embeddings were replaced with embeddings from a new text-only encoder (BAAI/bge-m3).** The corresponding kNN graph was computed using the new embeddings.
>
> ||EVQA||||InfoSeek||||
> |-|-|-|-|-|-|-|-|-|
> ||R@1|R@5|R@10|R@20|R@1|R@5|R@10|R@20|
> |Original|28.8|49.9|57.0|65.1|60.8|80.0|84.6|87.7|
> |BGE replacement|29.0|50.1|56.8|65.1|60.7|80.1|84.6|87.8|
>
> The results show that **retrieval performance remains stable.** These findings indicate that **the similarity signal is robust to the choice of encoders.** This observation supports the design assumption that SCAR can operate reliably even when different views are encoded using heterogeneous encoders while constructing kNN graphs.
>
> **Thank you for the constructive and technically rigorous feedback, which helped us improve the clarity and robustness of the work. We hope that the additional analyses have addressed your concerns.**
>
> [1]Chen, Thomas Y. "Rate-Distortion Limits for Multimodal Retrieval: Theory, Optimal Codes, and Finite-Sample Guarantees." Proceedings of the IEEE/CVF International Conference on Computer Vision. 2025.

---

> > ### Author Rebuttal · Reviewer_5yLT · 2026-04-03
> >
> > Most of my concerns are addressed. I will remain the score.

---

> > > ### Author Response · Authors · 2026-04-05
> > >
> > > Thank you again for your careful reassessment and for noting that most of your concerns have been addressed. We sincerely appreciate your time and thoughtful feedback throughout the discussion. We hope the revision has clarified the main technical points and addressed the key issues you raised, and that the rebuttal now better reflects the contribution of the work.

---

### Official Review · Reviewer_7Bdf · 2026-03-07

**Soundness:** 3
**Presentation:** 3
**Significance:** 3
**Originality:** 3
**Overall Recommendation:** 3
**Confidence:** 4

**Summary:**

This paper addresses the problem of Knowledge-Based Visual Question Answering (KB-VQA) , where a model must answer a question by retrieving relevant entities from a multimodal knowledge base   using both an image and a text query.   the paper proposes SCAR , a novel framework that performs retrieval over multiple entity views simultaneously  and then intelligently combines the results. The paper also provides a theoretical analysis to derive upper bounds on single-view and multi-view retrieval performance, explaining why multi-view approaches can improve the recall ceiling when views are complementary.

**Compliance With Llm Reviewing Policy:**

Affirmed.

**Key Questions For Authors:**

See the weaknesses.

**Limitations:**

See the weaknesses.

**Strengths And Weaknesses:**

Strengths:

S1: The paper clearly identifies a fundamental limitation of existing KB-VQA systems: reliance on a single entity representation leads to systematic blind spots. The proposed multi-view indexing is a principled solution to this problem, and the authors support it with both empirical evidence and theoretical analysis.

S2: The SCAR framework is technically sound and addresses the key challenges of multi-view retrieval: within-view noise and cross-view redundancy. The design choices are well-justified, and ablation studies confirm that both modules contribute positively. The method is also computationally efficient, adding only marginal overhead while delivering substantial recall gains.

Weaknesses:

W1: While the overall framework is novel, some of the individual techniques are borrowed from prior work. For examples, manifold diffusion for retrieval has been used in image retrieval. The paper could better highlight what is specifically new in its combination and adaptation of these ideas to the KB-VQA setting.

W2: While the paper shows overall gains, it does not systematically analyze where SCAR still fails. For example, when does the redundancy regulation mistakenly suppress a useful view? When does manifold diffusion over-smooth and merge distinct entities? A qualitative error analysis would provide deeper insights into limitations and guide future improvements.

---

> ### Author Rebuttal · Authors · 2026-03-31
>
> ## Request for clearer novelty announcement beyond combining known techniques.(W1)
>
> We do not simply combine existing techniques; instead, we introduce **a structural perspective** on retrieval in KB-VQA, derive a **principled design** from analysis, and demonstrate that proposed method can produce **synergistic gains beyond individual components**. We summarize the key novelty of our work across different levels below.
>
> |Level of Innovation|Novelty in Our Work|Comparison with Prior Work|
> |-|-|-|
> |New Empirical Insight|We identify **a key limitation of single-view retrieval in KB-VQA**, showing that different views provide complementary evidence beyond a single index.|Prior systems **rely on a single index for coarse retrieval** and focus on rerankers or MLLMs, without explicit multi-view coordination.(Zhang et al., 2024; Cocchi et al., 2025; Yang et al., 2025)|
> |New Analytical Perspective|We develop **a multi-view information-theoretic analysis** that characterizes bottlenecks in retrieval, including view-specific information and cross-view efficiency, revealing the potential of multi-view retrieval.|Existing analyses mainly study single-view behavior; **multi-view interactions are rarely formalized.**(Caffagni et al., 2024)|
> |New Method Design|We design **two complementary modules that enhance view-specific signals  and regulate cross-view redundancy**.|Manifold diffusion has been used in image retrieval, but **prior work typically applies it within a single representation space**, without addressing how view-specific noise should be controlled before multi-view aggregation.(Lin et al., 2023)|
> |New System Mechanism|We introduce **a parallel multi-view retrieval mechanism** that aggregates evidence from multiple indices while suppressing redundant signals in real time.|Prior pipelines **process a single index or apply components sequentially**, limiting coordination across views.(Yan & Xie, 2024)|
> |New Practical Outcome|The system achieves **substantial retrieval gains (up to 65.1% Recall@20 in E-VQA)** while remaining training-free and compatible with existing KB-VQA pipelines.|Many prior improvements rely on rerankers or model training, increasing deployment complexity. Meanwhile, coarse retrieval performance typically **remains around 55% Recall@20 in E-VQA.**(Yang et al., 2025b; Cocchi et al., 2025)|
>
> ## Request for systematic failure-case analysis.(W2)
>
> We agree that systematic qualitative analysis helps clarify the limitations of the method. Below we summarize representative failure conditions of the structural coordination mechanism.
>
> **Module I Failure: incorrect smoothing during manifold diffusion.** This occurs when the correct entity lies in a sparse neighborhood with weak similarity connections, while competing entities form a dense cluster. During diffusion, the dense cluster accumulates more propagated score, leaving the correct but isolated entity under-reinforced.
>
> **Module II Failure: informative view mistakenly suppressed.** This occurs when the correct entity appears in only one view, while other entities exhibit strong cross-view structural similarity. In this case, redundancy regulation may downweight the only informative view because its structure appears redundant.
>
> When these conditions occur, the correct entity receives limited gain from either module and may be overtaken by competing entities. The residual gap to the multi-view upper bound below provides an approximate empirical estimate of such failures.
>
> ||EVQA||||InfoSeek||||
> |-|-|-|-|-|-|-|-|-|
> ||R@1|R@5|R@10|R@20|R@1|R@5|R@10|R@20|
> |Upper Bound|33.9|56.9|65.8|73.0|68.9|84.2|87.9|90.7|
> |SOTA baseline|18.7|41.2|49.7|58.7|52.2|73.7|79.8|84.7|
> |Ours|28.8|49.9|57.0|65.1|60.8|80.0|84.6|87.7|
> |**Residual Gap to Upper Bound**|5.1|7.0|8.8|7.9|8.1|4.2|3.3|3.0|
>
> SCAR already preserves about 90% (E-VQA) and 97% (InfoSeek) of reachable correct entities, indicating that such failures are relatively infrequent. Representative examples of such failure cases are shown below.
>
> |Query ID|Retrieval Status|Failure Reason|
> |-|-|-|
> |EVQA-1003|Image hit (Rank 10), Summary/Title miss|Low-ranked correct entity in the image view, while **overlapping competitors from title/summary accumulate higher scores** through diffusion.|
> |EVQA-1014|Title hit (Rank 14), Image/Summary miss|Correct entity appears only in the title view, but **strong cross-view similarity leads to global downweighting during redundancy regulation.**|
> |EVQA-1068|Summary hit (Rank 17), Image/Title miss|Correct entity ranked **near the tail in the summary view, receiving limited reinforcement during diffusion.**|
>
> **Thank you for the constructive comments, which give us an opportunity to better articulate the novelty and practical behavior of the method. We hope your concerns have been addressed.**

---

> > ### Author Rebuttal · Reviewer_7Bdf · 2026-04-05
> >
> > Thanks for your responses. No more questions.

---

> > > ### Author Response · Authors · 2026-04-06
> > >
> > > Thank you for your review and for confirming that your concerns have been adequately addressed. We sincerely appreciate your time and the opportunity to clarify the work.
> > >
> > > If you feel that the rebuttal has satisfactorily resolved the earlier issues, we would be grateful if this could be reflected in the overall recommendation.

---

### Official Review · Reviewer_rbt8 · 2026-03-11

**Soundness:** 2
**Presentation:** 3
**Significance:** 3
**Originality:** 3
**Overall Recommendation:** 4
**Confidence:** 2

**Summary:**

This paper studies the retrieval stage in knowledge-based visual question answering (KB-VQA) and argues that existing systems rely too heavily on a single index view which often miss relevant knowledge. The authors show that different views are complementary. To address this, it proposes SCAR, a training-free multi-view retrieval framework that exploits structural relations within each view and reduces redundancy across views to better combine complementary evidence. Within each view, it builds an entity-level kNN graph and propagates similarity scores over that graph, so relevant entities can be reinforced by their structural neighbors rather than relying only on isolated query–entity similarity. Across views, it explicitly measures structural redundancy and suppresses overlapping evidence from lower-priority views, allowing the system to combine complementary signals while avoiding repeated amplification of the same information. Experiments on E-VQA and InfoSeek show that SCAR improves retrieval recall and downstream question-answering performance with minimal extra overhead.

**Compliance With Llm Reviewing Policy:**

Affirmed.

**Final Justification:**

I am maintaining my original positive rating.

**Key Questions For Authors:**

see Weaknesses

**Limitations:**

yes

**Strengths And Weaknesses:**

Strengths:

1、The paper focuses on coarse-grained retrieval in KB-VQA, a stage that is often underexplored compared with reranking or answering, so the problem setting is relevant and practically important.

2、The paper goes beyond single-view indexing and explicitly studies how different views can be combined, which is a reasonable and potentially useful direction for multimodal knowledge retrieval.

3、SCAR works at inference time without extra training, which makes it lightweight and appealing for existing KB-VQA pipelines.

4、The method shows consistent gains in retrieval recall and also improves downstream KB-VQA performance, suggesting that better coarse retrieval indeed benefits the whole pipeline.

5、The information-theoretic analysis helps frame why single-view retrieval may be limited and why multi-view retrieval could help.

Weaknesses:

1、Figure 1 only weakly supports the claim that “each view exhibits complementary strengths.” The Venn diagrams only show overlap in Recall@20, that is, whether different views retrieve the same entities. They demonstrate result-level complementarity, but do not directly prove that the observed gains come from the intrinsic strengths of each view. Figure 1 does not explain why a specific view retrieves certain unique entities, whether such unique hits are consistent for particular question types, or whether they arise from genuine semantic advantages rather than retrieval noise, index construction, or embedding bias. Therefore, it supports complementarity at the outcome level, but not necessarily “strengths” at the mechanism level.

2、The relatively low multi-view coverage upper bound in Table 1 suggests that the core challenge of KB-VQA retrieval may not be fully addressed by multi-view aggregation alone, but may instead stem from limited embedding quality and similarity estimation in coarse retrieval.

3、The method introduces a relatively large number of hyperparameters and initialization variables, making the overall pipeline complex and raising concerns about sensitivity, reproducibility, and tuning cost.

---

> ### Author Rebuttal · Authors · 2026-03-31
>
> ## Concern about lacking mechanism-level evidence for view strengths.(W1)
>
> We thank the reviewer for this insightful comment. We agree that **Figure 1 provides preliminary evidence illustrating coverage differences across views, and does not by itself establish intrinsic strengths at the mechanism level.** The purpose of Figure 1 is to **motivate the presence of systematic coverage gaps in single-view retrieval, rather than to serve as definitive proof of view-specific strengths.** Building on this observation, we develop SCAR that enhances view-specific signals while coordinating information across views. **We will revise the caption of Figure 1 and the related textual descriptions to more clearly position it as an observation of coverage differences.**
>
>
> ## Concern that embedding quality may be the main bottleneck.(W2)
>
> We agree that more powerful embeddings can significantly improve retrieval performance. **This observation is consistent with our theoretical analysis**, which shows that retrieval recall increases as the encoder captures more mutual information about the query and entity representations.
>
> **However, our work addresses a complementary dimension of the problem. In practice, current multimodal encoders (e.g., EVA-CLIP 8B), which are standard encoders used in KB-VQA tasks, still operate under finite representational capacity.** Our work therefore focuses on improving retrieval effectiveness under this regime, where the mutual information captured by the representation space is inherently finite. Under such conditions, better utilization of evidence across views becomes a critical factor for improving retrieval performance without changing the embedding.
>
> **Finally, our approach is fully compatible with future advances in embedding quality: as stronger embeddings become available, our method can directly benefit from them and achieve further gains.**
>
>
> ## Concern about hyperparameter burden and reproducibility.(W3)
>
> We thank the reviewer for raising this important point regarding parameter complexity and robustness.
>
> We agree that **sensitivity, reproducibility, and tuning cost are critical considerations for practical deployment.** To address these concerns, **we have conducted extensive parameter sensitivity analysis, as reported in the Appendix**.
>
> Specifically, we evaluate the impact of key hyperparameters, including the diffusion strength, similarity threshold, and overlap suppression factors. **As shown in Appendix Tables 14–18, retrieval performance remains stable across a wide range of values for each parameter**, indicating low sensitivity of the overall system behavior and demonstrating that **our method does not rely on careful parameter adjustment.**
>
> Importantly, all experiments in the paper use a fixed set of default parameters shared across datasets, without per-dataset tuning. This design ensures reproducibility and keeps the tuning cost minimal.
>
> **Thank you for the constructive and insightful feedback. Your comments helped us clarify the scope and strengthen the robustness of the system.**

---

> > ### Author Rebuttal · Reviewer_rbt8 · 2026-04-03
> >
> > The author answered my questions. Since I am not very familiar with this field, I am maintaining my original rating.

---

> > > ### Author Response · Authors · 2026-04-05
> > >
> > > Thank you very much for your careful reading and for confirming that your concerns have been fully resolved. We are encouraged that our responses helped clarify the questions you raised, and we hope the rebuttal now communicates the motivation, technical contribution, and empirical significance of the work more clearly.
> > >
> > > We sincerely appreciate your time and consideration.

---

### Official Review · Reviewer_wa2s · 2026-03-11

**Soundness:** 3
**Presentation:** 3
**Significance:** 2
**Originality:** 3
**Overall Recommendation:** 3
**Confidence:** 3

**Summary:**

This paper studies the coarse retrieval stage in knowledge-based-VQA. The authors argue that single-view retrieval is incomplete because different views of the same entity contain different information, and any one view can miss relevant evidence. To address this, they propose SCAR, a training-free multi-view retrieval method.

SCAR first builds a kNN graph over retrieved entities within each view (summary, image, and title) propagates retrieval scores over that graph, then compares graph structure across views to detect redundancy and downweight lower-priority views, and finally combines the view-specific scores into a single ranked entity list. Experiments on E-VQA and InfoSeek show improved Recall@K over the reported baselines, with some downstream gains on end-to-end VQA as well.

**Compliance With Llm Reviewing Policy:**

Affirmed.

**Final Justification:**

Given that the author clarify most of my concerns, I have increased the soundness score.

**Key Questions For Authors:**

1. Appendix A already notes that Assumption 1 is a technical simplification, and that dependence among distractors in modern embedding spaces should be viewed as structured noise rather than a faithful modeling assumption. Given that hard negatives in dense retrieval are often highly correlated, could the authors comment a bit more on how violations of this assumption may affect the interpretation of Eq. 6 in practice? Should the current theory mainly be read as a qualitative guide to the limiting factors of multi-view retrieval, rather than as a quantitatively tight characterization?

2. In the released code, when building the aligned entity–embedding matrix for the image view, the loop uses a simple overwrite (`full_matrix[idx] = vecs[i]`), and FAISS returns results in descending similarity order, the last occurrence for each entity—i.e., the lowest-similarity image among that entity’s hits—becomes its sole visual representation in the kNN graph and diffusion. Was this “keep last / worst hit” choice intentional (e.g., for a particular robustness or diversity reason), or would the authors consider using the best hit or an aggregation over multiple hits? If intentional, a short clarification in the paper or appendix would help readers and reproducers.

**Limitations:**

The cross-view regulation relies quite heavily on fixed rules. The method uses a hard-coded view priority order (Summary → Image → Title) together with fixed hyperparameters ((\tau), (\gamma), (\beta)). Although the paper shows through priority-permutation and sensitivity analyses that these choices are reasonably stable in the current experiments, the paper could be more direct about the boundary of this design: if the method is applied to a new domain with a missing modality, or with text and visual quality distributions that differ substantially from the current datasets, it is unclear whether the same non-adaptive fixed rules would remain reliable. In other words, the current results show that these heuristics work in the reported setting, but their robustness under changing view quality, missing modalities, or distribution shift has not yet been fully established.

(this one might not be super valid but is just something I was wondering->) The paper does not yet discuss in enough depth the gap between its theoretical assumptions and real dense retrieval. Appendix A assumes conditional independence among distractors given the query and target, but in real dense retrieval, the hardest cases are often exactly those hard negatives that cluster together and are semantically close to the target. The paper already notes that this is a technical simplification and is better read as a qualitative analysis, but that also means the current theory still has limited power to explain correlated noise, clustered distractors, and difficult edge cases in realistic semantic spaces.

**Strengths And Weaknesses:**

Strengths:

1. Clear research questions. The paper focus coarse retrieval in KB-VQA and argues that single-view retrieval has coverage gaps, while different views can provide complementary evidence. This motivation is clear and matches the patterns shown in Figure 1, Table 2, and Figure 6: the views do not cover the same candidates, and a single view can miss relevant entities. The paper also explains clearly that weak coarse retrieval limits later reranking and answering.

2. Method is clear at a high level. SCAR has two parts: intra-view similarity propagation on entity kNN graphs, and cross-view redundancy regulation based on structural similarity. Their roles are clear: the first uses local structure inside each view, and the second reduces redundant competition across views and avoids the diminishing returns of naive multi-view aggregation. The pipeline is easy to follow, and the method is training-free and inference-only, which is practically appealing.

3. The experiment gives reasonable empirical support. Table 1 shows that SCAR beats the reported retrieval baselines on E-VQA and InfoSeek. Table 2 shows that full three-view SCAR beats every single-view and two-view setting. Table 3 shows, on E-VQA, that SCAR still helps after reranking. Table 4 shows that retrieval gains also improve downstream KB-VQA. The paper also includes efficiency analysis, hyperparameter sensitivity. Authors tested the method from several angles.

That said, I still have several concerns.

1. The theory explains the method better than it guides the design. Section 4 and Appendix A introduce (M_{\mathrm{eff}}), (\rho), and (\kappa \Delta H) to explain multi-view gains and redundancy. This analysis is useful, but it feels more like a qualitative explanation than a strong theory that directly determines the algorithm. The appendix also relies on strong assumptions, such as distractor conditional independence, while hard-negative correlation is one of the main difficulties in dense retrieval. So the theory helps explain why the method may work, but it still feels somewhat removed from the actual setting.


2. Image-view entity representation is under-specified and may be suboptimal. In the released code, the image index has multiple FAISS rows per entity (one per image). When building the union and aligned matrices for SCAR, the code iterates over each view’s (ids, vecs) and does `full_matrix[idx] = vecs[i]` for each hit. FAISS returns results in descending similarity order, so when the same entity appears multiple times in the image top-k, the last occurrence (lowest similarity) overwrites earlier ones. Thus, the lowest-scoring image hit for that entity becomes its sole visual representation in the kNN graph and diffusion. This “last overwrite” rule is not documented in the paper or README. Using the worst among several candidate images as the single embedding for an entity seems a questionable design and could hurt retrieval; the paper should either disclose and justify this choice or change to a clearer strategy.

3. Baseline Comparisons. I do understand the proposed method is training-free. However, the paper could benefit from comparisons with stronger baselines, such as MuKA or PreFLMR, which achieve considerably better VQA performance on E-VQA and Infoseek. This would help better contextualize the gains from SCAR and show how much of the gap to state-of-the-art KBVQA systems.

---

> ### Author Rebuttal · Authors · 2026-03-31
>
> ## Concern about theory-to-method gap and simplifying assumptions.(W1&Q1)
>
>
> We clarify that **our analysis identifies key bottlenecks in multi-view retrieval rather than deriving a closed-form optimal algorithm.** Specifically, Section 4 identifies two limiting factors that directly inspire our design, summarized in the following Table.
>
> |Theory-Method Mapping|||||||||
> |-|-|-|-|-|-|-|-|-|
> |Perspective|⟶|Bottlenecks from theoretical analysis|⟶|Principles|⟶|Methods|⟶|Outcome|
> |In-View|⟶|Mutual Information $I$|⟶|Strengthen intra-view signal propagation|⟶|Similarity Manifold Diffusion|⟶|Recall@20 **improves from around 50 to 65.1**|
> |Cross-View|⟶|Effective number $M_{eff}$|⟶|Suppress structured redundancy to highlight complementary view signals|⟶|Cross-View Redundancy Regulation|⟶|Recalls approximate theoretical optimum **(~90% of the coverage upper bound)**|
>
> Regarding Assumption 1 and Eq. 6, **we clarify that the independence assumption is a technical simplification rather than a faithful model of embedding spaces.** In KB-VQA, each query is associated with one target entity, so distractor dependence mainly appears as hard negatives that increase ambiguity, rather than adding target-identifying information.
>
> Assumption 1 removes distractor identity information to derive a tractable bound focused on target-side factors. **When distractors are correlated, the bound is conservative rather than tight, while the qualitative dependence on limiting factors remains unchanged.** Accordingly, we position Eq. 6 as a qualitative guide to multi-view retrieval bottlenecks, not a quantitatively exact characterization.
>
>
> ## Concern about undocumented last-overwrite image aggregation.(W2&Q2)
>
>
> Thank you for pointing out this implementation detail. **We clarify that the current "last overwrite" rule for image-view representations is an implentation detail rather than an algorithmic design or a bug, which have minimal impact on the overall performance.** We have evaluated several alternative aggregation strategies, including selecting the highest-similarity embedding (first one), averaging multiple embeddings, and using the last retrieved embedding. **As shown below, these variants yield nearly identical performance**
>
> |Strategy|EVQA||||InfoSeek||||
> |-|:-:|-|-|-|:-:|-|-|-|
> ||R@1|R@5|R@10|R@20|R@1|R@5|R@10|R@20|
> |Average embedding|28.5|49.9|57.1|64.6|60.7|80.0|84.5|87.8|
> |First embedding|28.2|49.9|57.3|64.6|60.7|80.0|84.5|87.7|
> |Last embedding|28.8|49.9|57.0|65.1|60.8|80.0|84.6|87.7|
>
> This robustness is expected because the downstream diffusion and cross-view aggregation mechanisms depend primarily on neighborhood structure rather than the exact choice of a single image embedding. We will clarify this behavior in the revised paper and describe the aggregation policy explicitly for implmentation transparency.
>
>
> ## Request for stronger SOTA baseline comparisons.(W3)
>
>
> We thank the reviewer for suggesting comparisons with strong KB-VQA systems such as PreFLMR and MuKA. We would like to clarify that **the baselines included in our original submission (e.g., EchoSight, Reflectiva, and OMGM) already represent recent state-of-the-art KB-VQA systems.**  While works such as PreFLMR and MuKA also report results on E-VQA and InfoSeek, their original experimental protocols differ from the retrieval setting adopted in recent KB-VQA retrieval studies (e.g., differences in retrieval space, query distribution, and evaluation metrics). As a result, the originally reported numbers are not directly comparable to our setting.
>
> To enable the comparison, **we reproduced PreFLMR and MuKA under the same retrieval space, query set, and evaluation protocol.** The performance is summarized below.
>
> |Method|EVQA||||InfoSeek||||
> |-|:-:|-|-|-|:-:|-|-|-|
> ||R@1|R@5|R@10|R@20|R@1|R@5|R@10|R@20|
> |PreFLMR-ViT-G|14.6|25.7|31.0|36.3|15.4|34.3|43.2|51.0|
> |MuKA+PreFLMR|15.1|26.7|32.0|37.2|24.2|42.5|49.5|55.8|
> |SCAR (Ours)|28.8|49.9|57.0|65.1|60.8|80.0|84.6| 87.7|
>
> To verify the correctness of our reproductions, we additionally evaluated PreFLMR using the original metric. On InfoSeek, we obtained PR@5 = 55.52, which is close to the reported 59.6 in their paper. The remaining difference is expected because we use the full validation set (71,335 samples), while the original work reports results on a sampled subset (4,708 samples). For MuKA, the authors did not release model checkpoints (as confirmed in their public repository), so we reproduced the MuKA pipeline with the available PreFLMR weights.
>
> Overall, **these experiments provide a direct comparison to strong KB-VQA baselines** and further contextualize the performance gains of SCAR. We will include these results in the revised version of the paper to provide a more comprehensive evaluation against existing methods.
>
> **We would like to thank you for your conscientious review and valuable comments. Through the rebuttal process, both the clarity and the methodological details of our paper have been further improved.**

---

> > ### Author Rebuttal · Reviewer_wa2s · 2026-04-02
> >
> > Most of my concerns are addressed. I will increase the soundness score.

---

> > > ### Author Response · Authors · 2026-04-03
> > >
> > > Thank you again for your thoughtful reassessment and for acknowledging that the technical concerns have been resolved.
> > > We would also like to respectfully clarify the significance of our work.
> > >
> > > First, **the paper identifies an important but relatively underexplored bottleneck in KB-VQA**: coarse-grained retrieval fundamentally limits the downstream reranking and answering stages, yet prior work has largely focused on stronger rerankers or MLLMs rather than the retrieval ceiling itself. Our results show that this perspective is not only novel, but also highly effective in practice, leading to substantial improvements in coarse-grained retrieval recall.
> > >
> > > Second, **the empirical impact is consistent and practically meaningful**: SCAR substantially improves retrieval recall, approaches the multi-view coverage upper bound, and these gains further translate into stronger end-to-end KB-VQA performance, all with only negligible inference overhead. Importantly, the experiments show that significant improvements can be achieved under the same encoder setting, without relying on a stronger encoder or additional training, but purely through more effective exploitation of complementary information across views. We also note that **our gains are substantially larger than the relatively marginal differences among prior baselines**:
> > >
> > > ||EVQA||||InfoSeek||||
> > > |-|-|-|-|-|-|-|-|-|
> > > ||R@1|R@5|R@10|R@20|R@1|R@5|R@10|R@20|
> > > |Second Best baseline|18.9|36.8|46.2|55.6|49.7|71.6|78.0|82.5|
> > > |Best baseline|18.7|41.2|49.7|58.7|52.2|73.7|79.8|84.7|
> > > |Ours|28.8|49.9|57.0|65.1|60.8|80.0|84.6|87.7|
> > > |Best baseline - Second Best baseline|-0.2|+4.4|+3.5|+3.1|+2.5|+2.1|+1.8|+2.2|
> > > |**Ours - Best baseline**|**+10.1**|**+8.7**|**+7.3**|**+6.4**|**+8.6**|**+6.3**|**+4.8**|**+3.0**|
> > >
> > >
> > > Third, the results also suggest that **the method is particularly valuable in harder retrieval settings**. As reflected in the stronger gains on E-VQA, where the retrieval space is much larger (about 2 million entities) and single-view retrieval is more insufficient, SCAR appears especially effective when coarse-grained coverage is the key bottleneck. This suggests that the proposed multi-view retrieval strategy may be particularly relevant for more challenging multimodal retrieval problems beyond the specific benchmarks studied here.
> > >
> > > For these reasons, we hope the revision makes clearer that **the paper is not only technically sound, but also significant for multimodal KB-VQA, and potentially for broader multimodal retrieval settings. We would greatly appreciate your consideration of these points in the overall recommendation.**

---

### Decision · Program_Chairs · 2026-04-30

**Decision:**

Accept (regular)

**Comment:**

The paper studies coarse retrieval in KB-VQA and proposes a retrieval framework that combines information from summary, image, and title views. It received four reviews: two Weak Accepts and two Weak Rejects. The authors provided a detailed rebuttal, which led to a more positive overall discussion.

The reviewers agreed on several strengths (i) the paper addresses an important and underexplored problem in KB-VQA retrieval, where coarse retrieval is often overlooked compared to reranking and answer generation, (ii) the motivation for using multiple complementary views is clear and well supported, (iii) the proposed method is training-free, easy to integrate into existing systems, and adds little computational overhead, (iv) the method consistently improves both retrieval recall and downstream QA performance across datasets, and (v) ablations, studies

The reviewers noted some remaining concerns regarding the strength of the theoretical claims, the design choices and hyperparameters, and the limited analysis of certain failure cases and scalability. Post rebuttal, reviewers felt that the concerns were addressed mostly satisfactorily. The leftout concerns did not outweigh the strengths of the paper, as most were addressed in the rebuttal through additional experiments and clarifications.

The decision is to recommend acceptance.